# Reverse engineering of metacognition

**Matthias Guggenmos[1,2]\***

[1]Health and Medical University, Institute for Mind, Brain and Behavior, Potsdam, Germany; [2]Charité – Universitätsmedizin Berlin, Department of Psychiatry and Neurosciences, corporate member of Freie Universität Berlin and Humboldt-Universität zu Berlin, Berlin, Germany

**Abstract** The human ability to introspect on thoughts, perceptions or actions – metacognitive ability – has become a focal topic of both cognitive basic and clinical research. At the same time it has become increasingly clear that currently available quantitative tools are limited in their ability to make unconfounded inferences about metacognition. As a step forward, the present work introduces a comprehensive modeling framework of metacognition that allows for inferences about metacognitive noise and metacognitive biases during the readout of decision values or at the confidence reporting stage. The model assumes that confidence results from a continuous but noisy and potentially biased transformation of decision values, described by a confidence link function. A canonical set of metacognitive noise distributions is introduced which differ, amongst others, in their predictions about metacognitive sign flips of decision values. Successful recovery of model parameters is demonstrated, and the model is validated on an empirical data set. In particular, it is shown that metacognitive noise and bias parameters correlate with conventional behavioral measures. Crucially, in contrast to these conventional measures, metacognitive noise parameters inferred from the model are shown to be independent of performance. This work is accompanied by a toolbox (*ReMeta*) that allows researchers to estimate key parameters of metacognition in confidence datasets.

**\*For correspondence:**
mg.corresponding@gmail.com

**Competing interest:** The author declares that no competing interests exist.

## Editor's evaluation

This paper presents a novel computational model of metacognition and a validated toolbox for fitting it to empirical data. By formalizing different sources of noise and bias that impact confidence, the proposed model aims at providing metacognition metrics that are independent of perception – a continued endeavor in the field. The framework and toolbox constitute a valuable resource for the field.

## Introduction

The human ability to judge the quality of one's own choices, actions and percepts by means of confidence ratings has been subject to scientific inquiry since the dawn of empirical psychology (*Pierce and Jastrow, 1885*; *Fullerton and Cattell, 1892*), albeit it has long been limited to specific research niches. More recently, research on human confidence, and metacognition more generally, has accelerated and branched off to other domains such as mental illnesses (*Rouault et al., 2018*; *Hoven et al., 2019*; *Moritz and Lysaker, 2019*; *Seow et al., 2021*) and education (*Fleur et al., 2021*). Two main quantitative characteristics have emerged to describe subjective reports of confidence: *metacognitive bias* and *metacognitive sensitivity*.

*Fullerton and Cattell, 1892* already noted that 'different individuals place very different meanings on the degree of confidence. Some observers are nearly always quite or fairly confident, while others are seldom confident.' (p. 126). Technically, metacognitive biases describe a general propensity of

**eLife digest** Metacognition is a person's ability to think about their own thoughts. For example, imagine you are walking in a dark forest when you see an elongated object. You think it is a stick rather than a snake, but how sure are you? Reflecting on one's certainty about own thoughts or perceptions – confidence – is a prime example of metacognition. While our ability to think about our own thoughts in this way provides many, perhaps uniquely human, advantages, confidence judgements are prone to biases. Often, humans tend to be overconfident: we think we are right more often than we actually are. Internal noise of neural processes can also affect confidence.

Understanding these imperfections in metacognition could shed light on how humans think, but studying this phenomenon is challenging. Current methods are lacking either mechanistic insight about the sources of metacognitive biases and noise or rely on unrealistic assumptions. A better model for how metacognition works could provide a clearer picture.

Guggenmos developed a mathematical model and a computer toolbox to help researchers investigate how humans or animals estimate confidence in their own thoughts and resulting decisions . The model splits metacognition apart, allowing scientists to explore biases and sources of noise at different phases in the process. It takes two kinds of data: the decisions study participants make, and how sure they are about their decision being correct. It then recreates metacognition in three phases: the primary decision, the metacognitive readout of the evidence, and the confidence report. This allows investigators to see where and when noise and bias come into play. Guggenmos tested the model using independent data from a visual discrimination task and found that it was able to predict how confident participants reported to be in their decisions.

Metacognitive ability can change in people with mental illness. People with schizophrenia have often been found to be overconfident in their decisions, while people with depression can be underconfident. Using this model to separate the various facets of metacognition could help to explain why. It could also shed light on human thinking in general.

observers toward lower or higher confidence ratings, holding the accuracy of the primary actions – type 1 performance – constant. From a perspective of statistical confidence, that is assuming that observers use confidence ratings to report probability correct, an observer is often considered *underconfident* or *overconfident* if confidence ratings are systematically below or above the objective proportion of correct responses.

Metacognitive biases of this type have been quite extensively studied in the judgement and decision-making literature, in which they became known under the term *calibration* (*Lichtenstein et al., 1977b*). A central finding is that humans have a tendency toward overestimating their probability of being correct (*overconfidence bias*), particularly in general knowledge questions (*Lichtenstein et al., 1977b*; *Lichtenstein et al., 1982*; *Harvey, 1997*; but see *Gigerenzer et al., 1991*). More recently, overconfidence in decisions has been studied in psychiatric diseases, suggesting, for instance, underconfidence in individuals with depression (*Fu et al., 2005*; *Fu et al., 2012*; *Fieker et al., 2016*) and overconfidence in schizophrenic patients (*Moritz and Woodward, 2006a*; *Köther et al., 2012*; *Moritz et al., 2014*).

However, currently there is no established framework that allows for unbiased estimates of metacognitive biases. The validity of traditional calibration curve analyses, which is based on a comparison of the subjective and objective probability of being correct, has been debunked repeatedly (*Soll, 1996*; *Merkle, 2009*; *Drugowitsch, 2016*). In particular, the classic hard-easy (*Lichtenstein and Fischhoff, 1977a*), according to which overconfidence is particularly pronounced for difficult tasks, can be explained as a mere statistical artefact of random errors. For this reason, and in view of the potential importance in patient populations, there is a pressing need for unbiased measures of metacognitive biases.

While the measurement of metacognitive biases has received surprisingly little attention in the recent decades, the intricacies of measuring metacognitive sensitivity have been the subject of critical discussion and have spurred a number of methodological developments (*Nelson, 1984*; *Galvin et al., 2003*; *Maniscalco and Lau, 2012*; *Maniscalco and Lau, 2014*; *Fleming and Lau, 2014*). The issue is not the measurement of sensitivity per se: defining metacognitive (or type 2) sensitivity as the

ability to discriminate between one's correct and incorrect responses, it is readily possible to compute this quantity using the logic of receiver operating curve analyses (type 2 ROC; *Clarke et al., 1959*; *Pollack, 1959*). The main issue is that metacognitive sensitivity, according to this definition, is strongly influenced by type 1 performance. The lower type 1 performance, the higher will be the number of guessing trials and thus the higher will also be the expected number of trials in which observers assign low confidence to accidental correct guesses. Expected metacognitive sensitivity thus strongly depends on type 1 performance. Indeed, the importance of such type 1 performance confounds has been demonstrated in a recent meta-analysis of metacognitive performance aberrancies in schizophrenia (*Rouy et al., 2020*). The authors found that a previously claimed metacognitive deficit in schizophrenia was present only in studies that did not control for type 1 performance.

A potential solution to the problem of type 1 performance confounds was proposed by Maniscalco and colleagues through a measure called *meta-d'* (*Rounis et al., 2010*; *Maniscalco and Lau, 2012*; *Maniscalco and Lau, 2014*). Since *meta-d'* is expressed in units of d', it can be directly compared to – and normalized by – type 1 sensitivity, leading to a ratio measure termed $M_{ratio}$ ($M_{ratio}$ = *meta-d'* / d').

Recently, however, these normalized measures have come under scrutiny. *Bang et al., 2019* showed that the type 1 performance independence of $M_{ratio}$ breaks down with the simple assumption of a source of metacognitive noise that is independent of sensory noise. *Guggenmos, 2021* confirmed this diagnosis in a systematic analysis of empirical (Confidence Database; *Rahnev et al., 2020*) and simulated data. The very same factor (metacognitive noise) that therefore plausibly introduces interindividual differences in metacognitive performance, might obviate a type-1-performance-independent measurement of metacognitive efficiency in this way. Apart from type 1 performance, a recent study has shown that in empirical data the *overall level of confidence* likewise affects $M_{ratio}$ (*Xue et al., 2021*) – a confound that may be caused by different levels of metacognitive noise when overall confidence is low or high (*Shekhar and Rahnev, 2021*).

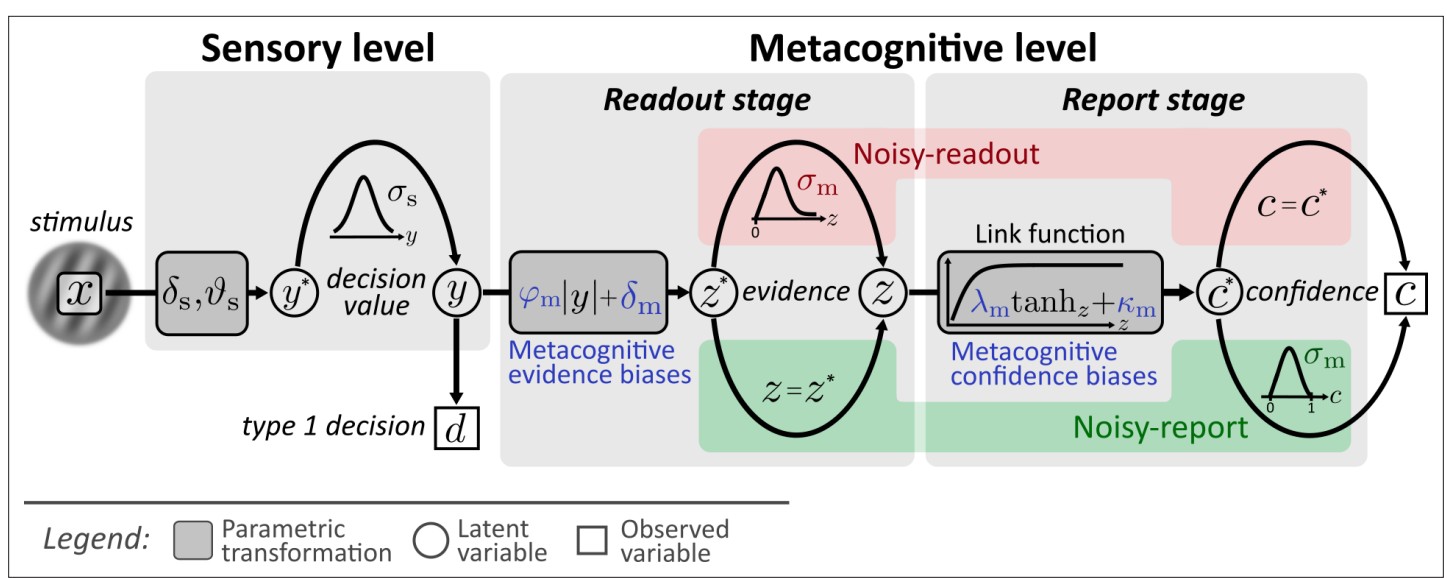

**Figure 1.** Computational model. Input to the model is the stimulus variable x, which codes the stimulus category (sign) and the intensity (absolute value). Type 1 decision-making is controlled by the sensory level. The processing of stimuli x at the sensory level is described by means of sensory noise ($\sigma_s$), bias ($\delta_s$) and threshold ($\vartheta_s$) parameters. The output of the sensory level is the decision value y, which determines type 1 decisions d and provides the input to the metacognitive level. At the metacognitive level it is assumed that the dominant source of metacognitive noise is either noise at the readout of decision values (*noisy-readout model*) or at the reporting stage (*noisy-report model*). In both cases, metacognitive judgements are based on the absolute decision value |y| (referred to as *sensory evidence*), leading to a representation of *metacognitive evidence* $z^*$ at the metacognitive level. While the "readout" of this decision value is considered precise for the noisy-report model ($z = z^*$), it is subject to metacognitive readout noise $z \sim f_m(z; z^*, \sigma_m)$ in the noisy-readout model, described by a metacognitive noise parameter $\sigma_m$. A link function transforms metacognitive evidence to internal confidence $c^*$. In the case of a noisy-report model, the dominant metacognitive noise source is during the report of confidence, that is confidence reports c are noisy expressions of the internal confidence representation: $c \sim f_m(c; c^*, \sigma_m)$. Metacognitive biases operate at the level of sensory evidence (*multiplicative evidence bias $\varphi_m$, additive evidence bias $\delta_m$*) or at the level of the confidence link function (*multiplicative confidence bias $\lambda_m$, additive confidence bias $\kappa_m$*).

Here I argue that an unbiased estimation of latent metacognitive parameters requires a mechanistic forward model – a process model which specifies the transformation from stimulus input to the computations underlying confidence reports and which considers sources of metacognitive noise. In the current work, I introduce a model and a toolbox to realize a process model approach for typical confidence datasets. It allows researchers to make parametric inferences about metacognitive inefficiencies either during readout or during report, as well as about different types of metacognitive biases. The basic structure of the model is shown in *Figure 1*. It comprises two distinct levels for type 1 decision making (*sensory level*) and type 2 metacognitive judgments (*metacognitive level*).

A few key design choices deserve emphasis. First, the model assumes that confidence is a second-order process (*Fleming and Daw, 2017*) which assesses the evidence that guided type 1 behavior. In the proposed nomenclature of *Maniscalco and Lau, 2016* it corresponds to a hierarchical model and not to a single-channel model in that it considers additional sources of metacognitive noise. A consequence of the hierarchical structure is that it is essential to capture the processes underlying the decision values at the type 1 level as precisely as possible, since decision values are the input to metacognitive computations. In the present model, this includes an estimate of both a sensory bias and a sensory threshold, both of which will influence type 1 decision values.

Second, recent work has demonstrated that metacognitive judgements are not only influenced by sensory noise, but also by metacognitive noise (*Bang et al., 2019*; *Shekhar and Rahnev, 2021*). In the present model, I therefore consider sources of metacognitive noise either during the readout of type 1 decision values or during report.

Third, human confidence ratings are often subject to metacognitive biases which can lead to the diagnosis of underconfidence or overconfidence. As outlined above, there is currently no established methodology to measure under- and overconfidence, let alone measure different types of such biases. In the present model, I consider four parameters that can be interpreted as metacognitive biases either at the level of evidence or at the level of the confidence report. The interpretation of these parameters as metacognitive biases entails the assumption that observers aim at reporting probability correct with their confidence ratings (*statistical confidence*; *Hangya et al., 2016*). Although I discuss link functions that deviate from this assumption, in the model outlined here, the transformation of sensory evidence to confidence therefore follows the logic of statistical confidence.

I demonstrate the issues of conventional measures of metacognitive ability and metacognitive biases, in particular their dependency on type 1 performance, and show that the process model approach can lead to unbiased inferences. Finally, I validate the model on a recently published empirical dataset (*Shekhar and Rahnev, 2021*). I illustrate for this dataset how model parameters can describe different facets of metacognition and assess the relationship of these parameters to conventional measures of metacognitive ability and metacognitive bias.

This article is accompanied by a toolbox – the Reverse engineering of Metacognition (*ReMeta*) toolbox, which allows researchers to apply the model to standard psychophysical datasets and make inferences about the parameters of the model. It is available at https://github.com/m-guggenmos/remeta, (copy archived at swh:1:rev:43ccbf2e35b1e934dab83e156e4fbb22ac160cd2; *Guggenmos, 2022*).

## Results

Results are structured in three parts. The first part introduces the architecture and the computational model, from stimulus input to type 1 and type 2 responses. The second part provides the mathematical basis for model inversion and parameter fitting and systematically assesses the success of parameter recovery as a function of sample size and varied ground truth parameter values. Finally, in the third part, the model is validated on an empirical dataset (*Shekhar and Rahnev, 2021*).

### Computational model
### Computing decision values

For the model outlined here, the task space is restricted to two stimulus categories referred to as $S^-$ and $S^+$. Stimuli are described by the stimulus variable $x$, the sign of which codes the stimulus category and the absolute value $|x|$ codes the intensity of the stimulus. The sensory level computes *decision values* $y^*$ from the stimulus input $x$ as follows:

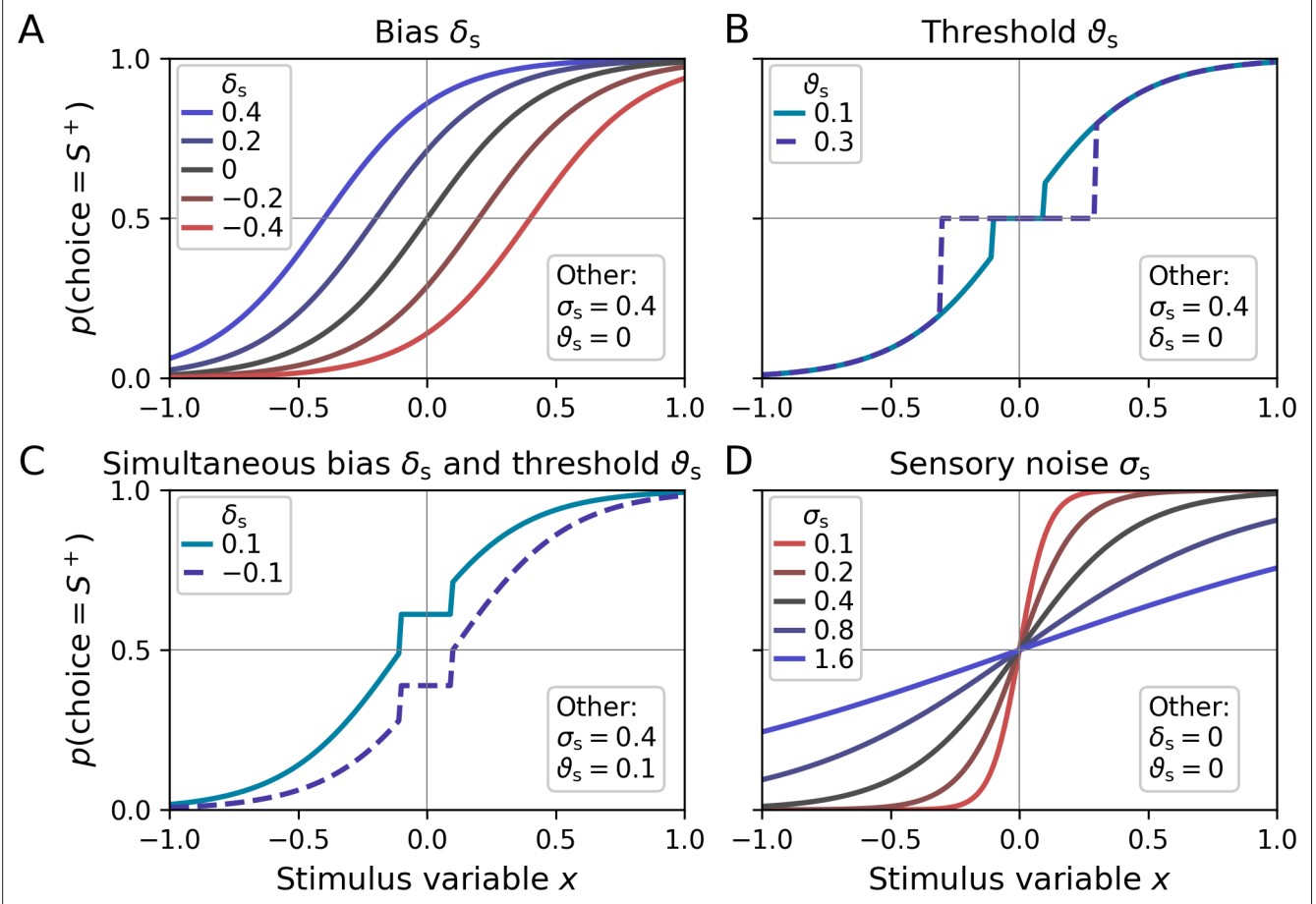

**Figure 2.** Psychometric functions for different settings of sensory model parameters. Top left legends indicate the values of varied parameters, bottom right legends settings of the respective other parameters. (**A**) The sensory bias parameter $\delta_s$ horizontally shifts the psychometric function, leading to a propensity to choose stimulus category $S^-$ ($\delta_s < 0$) or stimulus category $S^+$ ($\delta_s > 0$). (**B**) Stimulus intensities below the threshold parameter $\vartheta_s$ lead to chance-level performance. (**C**) Example for simultaneous non-zero values of the bias and threshold parameter. (**D**) The sensory noise parameter $\sigma_s$ changes the slope of the psychometric function.

The online version of this article includes the following figure supplement(s) for figure 2:

**Figure supplement 1.** Nonlinear transformation of the stimulus variable.

$$y^* = \begin{cases} x + \delta_s & \text{if } |x| > \vartheta_s \\ \delta_s & \text{else} \end{cases} \tag{1}$$

The sensory bias parameter $\delta_s \epsilon \mathbb{R}$ captures systematic preferences for one response category (**Figure 2A**) and corresponds to a horizontal shift of the resulting psychometric function. Positive (negative) values of $\delta_s$ lead to a propensity to choose stimulus category $S^+$ ($S^-$). In addition, the sensory threshold $\vartheta_s \epsilon \mathbb{R}^+$ defines the minimal stimulus intensity which is necessary to drive the system, that is, above which the observer's type 1 choices can be better than chance level (**Figure 2B**). Decision values $y^*$ are fixed to zero below $\vartheta_s$ in the absence of a sensory bias, and fixed to $\delta_s$ in the presence of a bias (**Figure 2C**). Note that a sensory threshold parameter should only be considered if the stimulus material includes intensity levels in a range at which participants perform close to chance. Otherwise, the parameter cannot be estimated and should be omitted, that is, **Equation 1** reduces to $y^* = x + \delta_s$.

In the model described here I assume that decision values can be linearly constructed from the stimulus variable x. In practice, this may often be too strong of an assumption, and it may thus be necessary to allow for a nonlinear transformation of x ('nonlinear transduction', see e.g. **Dosher and Lu, 1998**). The toolbox therefore offers an additional nonlinear transformation parameter $\gamma_s$ (see **Figure 2—figure supplement 1** for an illustration).

The final decision value $y$ is subject to sources of sensory noise $\sigma_s$, described by a logistic distribution $f_s(y)$:

$$y \sim f_s(y) = \frac{\pi}{\sqrt{3}\sigma_s} \frac{\exp\left(\frac{\pi(y-y^*)}{\sqrt{3}\sigma_s}\right)}{\left(1+\exp\left(\frac{\pi(y-y^*)}{\sqrt{3}\sigma_s}\right)\right)^2} \qquad (2)$$

*Equation 2* is a reparameterization of a standard logistic distribution in terms of the standard deviation $\sigma_s$ using the fact that the standard deviation of the logistic distribution is equal to $s\pi/\sqrt{3}$ (where $s$ is the conventional scale parameter of the logistic distribution). *Figure 2D* shows psychometric functions with varying levels of sensory noise $\sigma_s$. The logistic distribution was chosen over the more conventional normal distribution due to its explicit analytic solution of the cumulative density – the logistic function. In practice, both distributions are highly similar, and which one is chosen is unlikely to matter.

Type 1 decisions $d$ between the stimulus categories $S^+$ and $S^-$ are based on the sign of $y$:

$$d = \begin{cases} S^+ & \text{if } y \geq 0 \\ S^- & \text{if } y < 0 \end{cases} \qquad (3)$$

## From decision values to metacognitive evidence

The decision values computed at the sensory level constitute the input to the metacognitive level. I assume that metacognition leverages the same sensory information that also guides type 1 decisions (or a noisy version thereof). Specifically, metacognitive judgements are based on a readout of absolute decision values $|y|$, henceforth referred to as *sensory evidence*. Respecting a multiplicative ($\varphi_m \in R^+$) and an additive ($\delta_m \in R$) evidence bias, an estimate of sensory evidence is computed at the metacognitive level – *metacognitive evidence* $z^*$:

$$z^* = \max\left(\varphi_m |y| + \delta_m, \, 0\right) \qquad (4)$$

The multiplicative evidence bias $\varphi_m$ and the additive evidence bias $\delta_m$ are two different types of metacognitive biases at the readout stage, which are described in more detail in 'Metacognitive biases'. Note that the *max* operation is necessary to enforce positive values of metacognitive evidence.

## The link function: from metacognitive evidence to confidence

The transformation from metacognitive evidence to internal confidence $c^*$ is described by a *link function*. A suitable link function must be bounded, reflecting the fact that confidence ratings typically have lower and upper bounds, and increase monotonically.

I assume that observers aim at reporting probability correct, leading to a logistic link function in the case of the logistic sensory noise distribution (*Equation 2*). Without loss of generality, I use the range [0;1] for confidence ratings, such that a confidence level of 0 indicates expected chance-level type 1 performance (probability correct = 0.5) and a confidence level of 1 the expectation of optimal type 1 performance (probability correct = 1.0). Note that I do not consider the possibility that type 1 errors can be reported at the time of the confidence report, that is, confidence cannot be negative. With these constraints and using the simple mathematical relationship between the logistic function and the tangens hyperbolicus, one arrives at the following link function (see Appendix 1, *Equation A1*, for the derivation):

$$c^* = \tanh\left(\frac{\pi}{2\sqrt{3}\sigma_s} z\right) \qquad (5)$$

Note that I use the variable $z$ as opposed to $z^*$, to indicate that the metacognitive evidence that enters the link function may be a noisy version of $z^*$ (see the description of the *noisy-readout model* below). *Figure 3* shows examples of evidence-confidence relationships based on the link function in *Equation 5* and in dependence of several model parameters.

Many other link functions are conceivable, which do not assume that observers aim at expressing confidence as probability correct. In particular, such link functions may not involve an estimate of

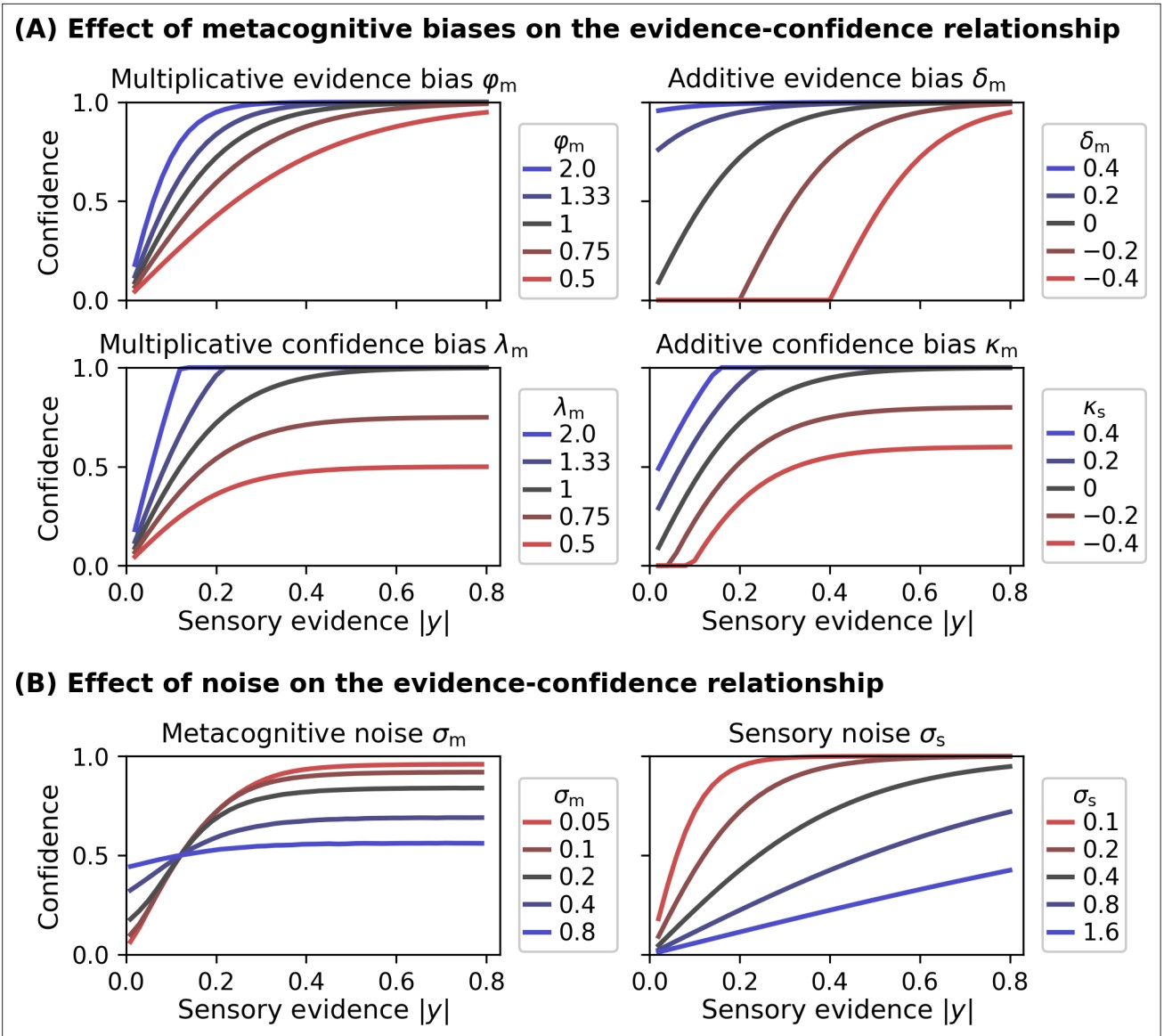

**Figure 3.** Effect of model parameters on the evidence-confidence relationship. All metacognitive bias parameters and noise parameters affect the relationship between the sensory evidence $|y|$ and confidence, assuming the link function provided in *Equation 5*. (**A**) Effect of metacognitive bias parameters on the evidence-confidence relationship. Metacognitive noise was set to zero for simplicity. (**B**) Effect of metacognitive noise $\sigma_m$ and sensory noise $\sigma_s$ on the evidence-confidence relationship. Metacognitive noise renders confidence ratings more indifferent with respect to the level of sensory evidence. Note that, due to the absence of an analytic expression, the illustration for the effect of metacognitive noise is based on simulation. Increasing sensory noise affects the slope of the confidence-evidence relationship, reflecting changes to be expected from an ideal metacognitive observer.

The online version of this article includes the following figure supplement(s) for figure 3:

**Figure supplement 1.** Confidence link functions.

sensory noise $\sigma_s$. *Figure 3—figure supplement 1* illustrates alternative link functions provided by the *ReMeta* toolbox.

I refer to $c^*$ as the *internal confidence*, which may be different from the ultimately *reported confidence* $c$. This distinction becomes important when metacognitive noise is considered at the level of the confidence report (see Result, 'Metacognitive noise: noisy-report models').

## Metacognitive biases

Metacognitive biases describe a systematic discrepancy between objective type 1 performance and subjective beliefs thereof (expressed via confidence ratings). Relative to an ideal metacognitive

observer of stastical confidence, overconfident observers report systematically higher levels of confidence and underconfident observers report systematically lower levels of confidence. Importantly, metacognitive biases are orthogonal to the metacognitive *sensitivity* of an observer. For instance, an underconfident observer who consistently chooses the second-lowest confidence rating for correct choices could have high metacognitive sensitivity nevertheless, as long as they consistently choose the lowest rating for incorrect choices. In the present model I consider metacognitive biases either at the level of evidence or at the level of confidence (*Figure 1*).

Metacognitive evidence biases represent a biased representation of sensory evidence at the metacognitive level. These biases may be either due to a biased readout from sensory channels or due to biased processing of read-out decision values at the initial stages of the metacognitive level. In either case, evidence biases affect the metacognitive representation $z$ of sensory evidence and may be multiplicative or additive in nature. The *multiplicative evidence bias* $\varphi_m$ leads to a scaling of absolute sensory decision values, with $\varphi_m < 1$ and $\varphi_m > 1$ corresponding to under- and overconfident observers, respectively. The *additive evidence bias* $\delta_m$ represents an additive bias such that metacognitive evidence is systematically decreased (underconfidence) or increased (overconfidence) by a constant $\delta_m$. Values $\delta_m < 0$ can be interpreted as a metacognitive threshold, such that the metacognitive level is only 'aware' of stimuli that yield sensory evidence above $\delta_m$.

An alternative interpretation of metacognitive evidence biases at the readout stage is that they correspond to an under- or overestimation of one's own sensory noise $\sigma_s$. Applying this view, a value of $\varphi_m > 1$ would suggest that the observer underestimated sensory noise $\sigma_s$ and hence shows overconfidence, whereas a value of $\varphi_m < 1$ implies that the observer overestimated $\sigma_s$ and thus is underconfident.

In addition, the present model considers metacognitive bias parameters loading on internal confidence representations. To this end, the confidence link function (*Equation 5*) is augmented by a *multiplicative confidence bias* $\lambda_m$ and an *additive confidence bias* $\kappa_m$:

$$c^* = \lambda_m \tanh\left(\frac{\pi}{2\sqrt{3}\sigma_s}z\right) + \kappa_m \tag{6}$$

Analogous to the evidence biases, values of $\lambda_m < 1$ and $\kappa_m < 0$ reflect underconfidence, and values of $\lambda_m > 1$ and $\kappa_m > 0$ reflect overconfidence. The effects of all metacognitive evidence and confidence bias parameters are illustrated in *Figure 3A*.

To assess how evidence- and confidence-related metacognitive biases relate to conventional measures of under- and overconfidence, I computed calibration curves (*Lichtenstein et al., 1977b*) for a range of values for each bias parameter (*Figure 4*, left panels). A first observation concerns the case in which no metacognitive biases are present (i.e. $\varphi_m = \lambda_m = 1$, $\delta_m = \kappa_m = 0$; black lines). One could assume that calibration curves for bias-free observers are identical to the diagonal, such that objective and subjective accuracy are identical. This is not the case – the calibration curve is tilted toward overconfidence. This may seem surprising but reflects exactly what is expected for a bias-free statistical confidence observer. This is best understood for the extreme case when the subjective probability correct is arbitrarily close to 1. Even for very high ratings of subjective probability, due to sensory noise, there is a certain finite probability that associated type 1 choices have been incorrect. Hence, objective type 1 performance is expected to be below the subjective probability in these cases. Importantly, *relative* to this bias-free observer all metacognitive bias parameters yield calibration curves that resemble under- and overconfidence given appropriate choices of the parameter values (underconfidence: redhish lines; overconfidence: blueish lines).

As mentioned previously, metacognitive sensitivity (*AUROC2*, *meta-d'*) is strongly dependent on type 1 performance. How do metacognitive biases perform in this regard, when measured in a model-free manner from choice and confidence reports? To find out, I simulated confidence biases for a range of metacognitive bias parameter values and type 1 performance levels (by varying the sensory noise parameter). Confidence biases were computed as the difference between subjective probability correct (by linearly transforming confidence from rating space [0; 1] to probability space [0.5; 1]) and objective probability correct. As shown in the middle panels of *Figure 4*, these results showcase the limits of naively measuring confidence biases in this way. Again, the bias-free observer shows an apparent overconfidence bias. In addition, this bias increases as type 1 performance decreases, reminiscent of the classic hard-easy effect for confidence (*Lichtenstein and Fischhoff, 1977a*; for related analyses, see *Soll, 1996*; *Merkle, 2009*; *Drugowitsch, 2016*; *Khalvati et al., 2021*). At chance level performance, the overconfidence bias is exactly 0.25.

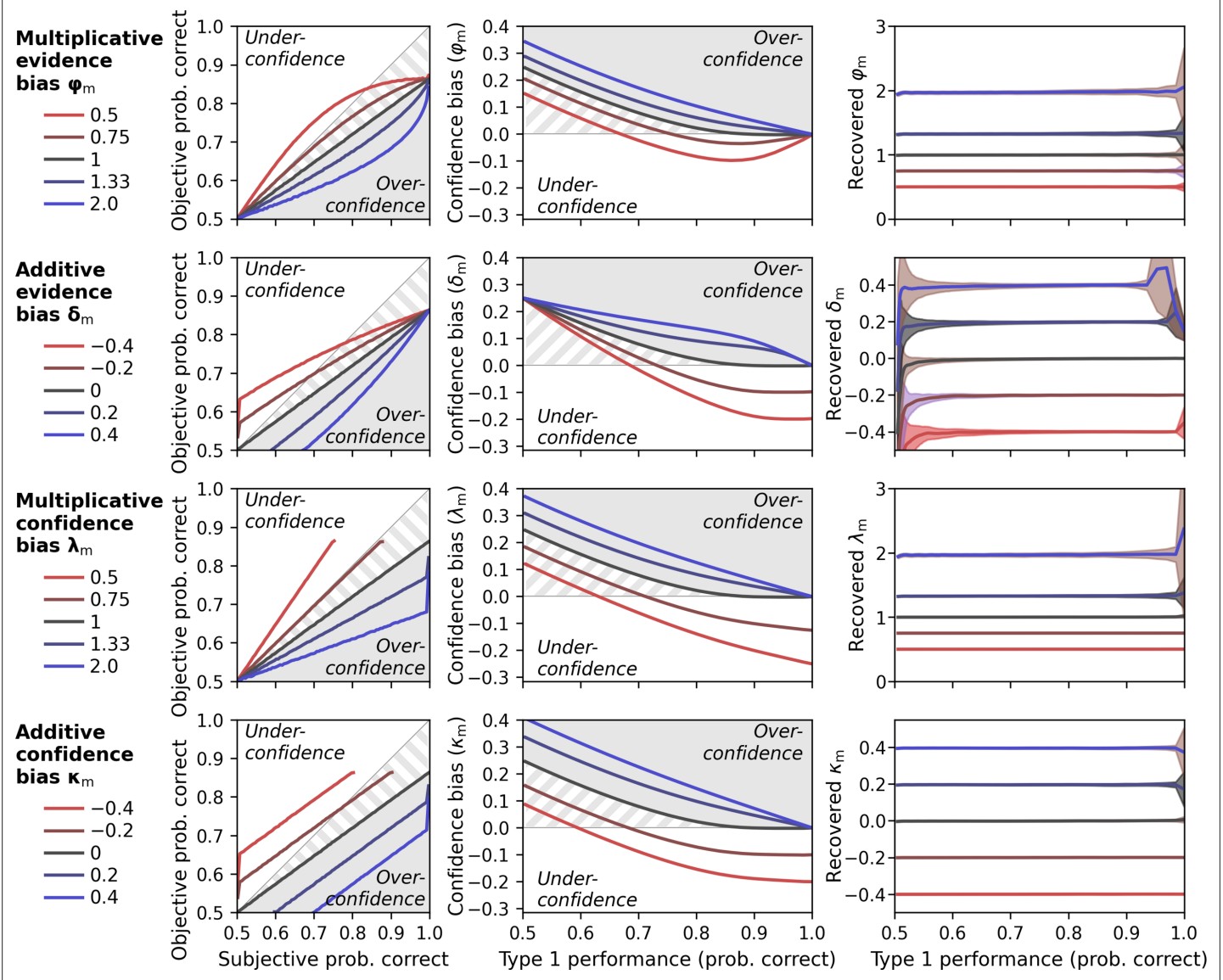

**Figure 4.** Metacognitive bias parameters ($\varphi_m$, $\delta_m$, $\lambda_m$, $\kappa_m$). Gray shades indicate areas of true overconfidence according to the generative model. Gray stripes areas indicate additional areas that would be classified as overconfidence in conventional analyses of confidence data, i.e. when simply comparing objective und subjective probability correct. Simulations are based on a noisy-report model with a truncated normal metacognitive noise distribution. Metacognitive noise was set close to zero for simplicity. (**Left panels**) Calibration curves. Calibration curves compute the proportion of correct responses (objective probability correct) for each interval of subjective confidence reports. Calibration curves above and below the diagonal indicate under- and overconfident observers, respectively. For this analysis, confidence was transformed from rating space [0; 1] to probability space [0.5; 1] and divided in 100 intervals with bin size 0.01. Average type 1 performance for this simulation was around 70%. (**Middle panels**) Confidence bias in dependence of type 1 performance. Different levels of type 1 performance were simulated by sweeping the sensory noise parameter between 0.01 and 50. Confidence bias was computed as the difference between subjective probability correct and objective proportion correct. (**Right panels**) Recovery of metacognitive bias parameters in dependence of performance. Shades indicate standard deviations.

The value of 0.25 can be understood in the context of the '0.75 signature' (*Hangya et al., 2016*; *Adler and Ma, 2018b*). When evidence discriminability is zero, an ideal Bayesian metacognitive observer will show an average confidence of 0.75 and thus an apparent (over)confidence bias of 0.25. Intuitively this can be understood from the fact that Bayesian confidence is defined as the area under a probability density in favor of the chosen option. Even in the case of zero evidence discriminability, this area will always be at least 0.5 – otherwise the other choice option would have been selected, but often higher.

The overconfidence bias leads to another peculiar case, namely that the bias of truly underconfident observers (i.e. $\varphi_m < 1$, $\delta_m < 0$, $\lambda_m < 1$, or $\kappa_m < 0$) can show a sign flip from over- to underconfidence as performance increases from chance level to perfect performance (redish lines in the middle panels of *Figure 4*). Overall, the simulation underscores that metacognitive biases are just as confounded by type 1 behavior as metacognitive sensitivity.

Is it possible to recover unbiased estimates for the metacognitive bias parameters by inverting the process model? To find out, I again simulated data for a range of type 1 performance levels and true values of the bias parameters. In each case, I fitted the model to the data to obtain estimates of the parameters. As shown in the right panels of *Figure 4*, parameter recovery was indeed unbiased across the type 1 performance spectrum, with certain deviations only for extremely low or high type 1 performance levels. This demonstrates that, in principle, unbiased inferences about metacognitive biases are possible in a process model approach, assuming that the fitted model is a sufficient approximation of the empirical generative model.

Finally, note that the parameter recovery shown in *Figure 4* was performed with four separate models, each of which was specified with a single metacognitive bias parameter (i.e., $\varphi_m$, $\delta_m$, $\lambda_m$, or $\kappa_m$). Parameter recovery can become unreliable when more than two of these bias parameters are specified in parallel (see 'Parameter recovery'). In practice, the researcher thus must make an informed decision about which bias parameters to include in a specific model. In most scenarios one or two metacognitive bias parameters are likely a good choice. While the evidence-related bias parameters $\varphi_m$ and $\delta_m$ have a more principled interpretation (e.g. as an under/overestimation of sensory noise), it is not unlikely that metacognitive biases also emerge at the level of the confidence report ($\lambda_m$, $\kappa_m$). The first step thus must always be a process of model specification or a statistical comparison of candidate models to determine the final specification (see also 'On using the model framework').

## Confidence criteria

In the model outlined here, confidence results from a continuous transformation of metacognitive evidence, described by a parametric link function (*Equation 5*). The model thus has no confidence criteria. However, it would be readily possible to replace the tangens hyperbolicus with a stepwise link function where each step is described by the criterion placed along the z-axis and the respective confidence level (alternatively, one can assume equidistant confidence levels, thereby saving half of the parameters). Such a link function might be particularly relevant for discrete confidence rating scales where participants associate available confidence ratings with often idiosyncratic and not easily parameterizable levels of metacognitive evidence.

Yet, even for the parametric link function of a statistical confidence observer it is worth considering two special confidence criteria: a minimum confidence criterion, below which confidence is 0, and a maximum criterion, above which confidence is 1. Indeed, the over-proportional presence of the most extreme confidence ratings that is often observed in confidence datasets (Confidence Database; *Rahnev et al., 2020*) motivates such criteria.

My premise here is that these two specific criteria can be described as an implicit result of metacognitive biases. In general, when considering an ideal statistical confidence observer and assuming continuous confidence ratings, the presence of any criterion reflects suboptimal metacognitive behavior – including a minimum or maximum confidence criterion. According to *Equation 5*, an ideal observer's confidence should never be exactly 1 (for finite sensory noise) and should only ever be 0 when metacognitive evidence is exactly zero, which makes a dedicated criterion for this case likewise superfluous.

Importantly, a minimum confidence criterion is implicit to the additive evidence bias $\delta_m$. As explained above, a negative value of $\delta_m$ effectively corresponds to a metacognitive threshold, such that metacognitive evidence $z$ (and hence confidence) is zero for decision values smaller than $\delta_m$. A maximum confidence criterion can be realized by the confidence bias parameters $\lambda_m$ and $\kappa_m$. Specifically, assuming $\lambda_m > 1$ or $\kappa_m > 0$, the maximum criterion is the point along the metacognitive evidence axis at which a link function of the form $\lambda_m \cdot \tanh(..) + \kappa_m$ becomes exactly 1. In sum, both a minimum and a maximum confidence criterion can be implemented as a form of a metacognitive bias.

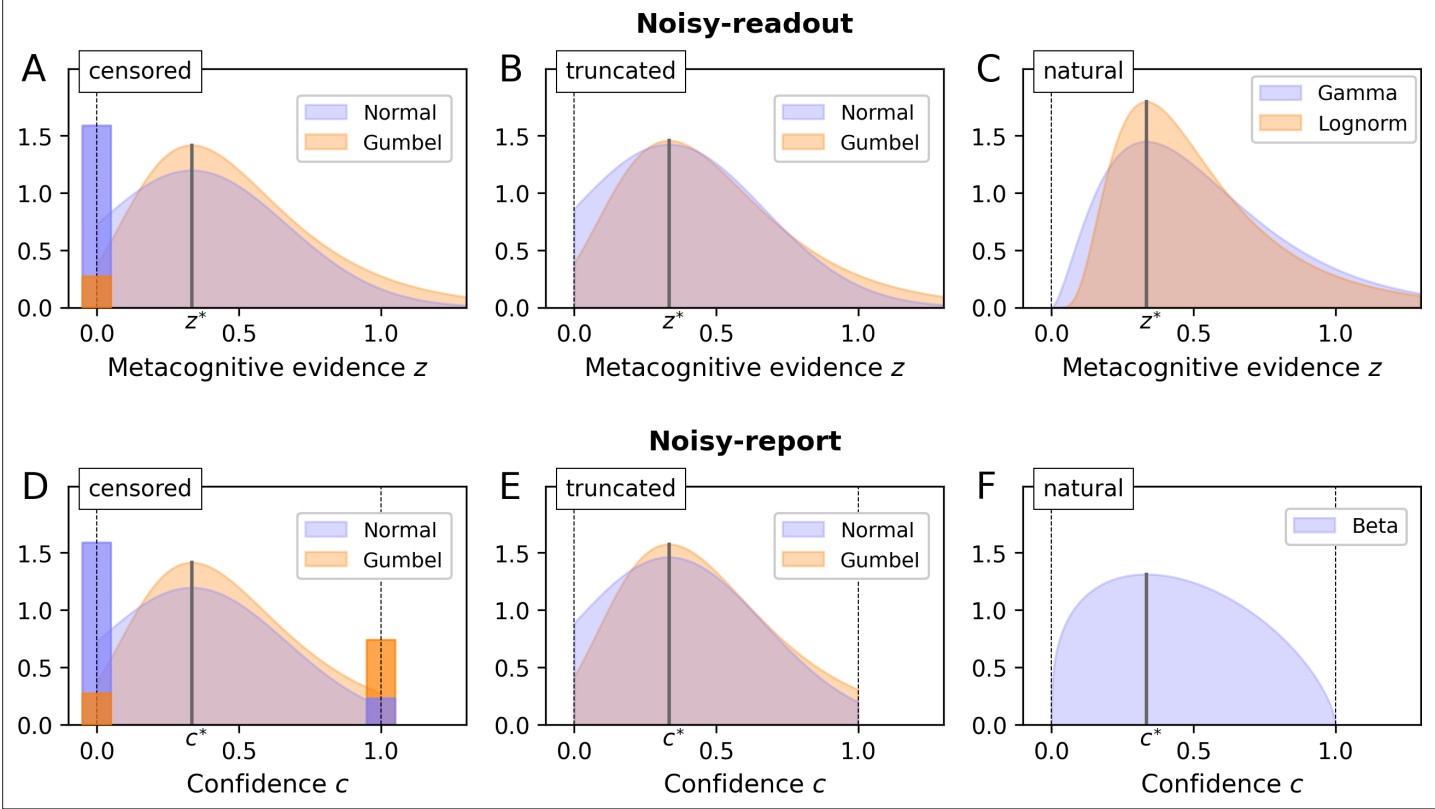

**Figure 5.** Metacognitive noise. Considered noise distributions are either censored, truncated or naturally bounded. In case of censoring, protruding probability mass accumulates at the bounds (depicted as bars with a darker shade; the width of these bars was chosen such that the area corresponds to the probability mass). The parameter $\sigma_m$ and the distributional mode was set to ⅓ in all cases (arbitrary value). (**A - C**) Noisy-readout models. Metacognitive noise is considered at the level of readout, affecting metacognitive evidence $z$. Only a lower bound at $z = 0$ applies. (**D - F**) Noisy-report models. Metacognitive noise is considered at the level of the confidence report, affecting internal confidence representations $c$. Confidence reports are bounded between 0 and 1.

## Metacognitive noise: noisy-readout models

A key aspect of the current model is that the transformation from sensory decision values to confidence reports is subject to sources of metacognitive noise. In this section, I first consider a model of type *noisy-readout*, according to which the metacognitive noise mainly applies to the metacognitive readout of absolute sensory decision values (i.e. $z^*$). The final metacognitive evidence $z$ is thus a noisy version of $z^*$. By contrast, sources of noise involved in the report of confidence are considered negligible and the internal confidence estimate $c^*$ resulting from the link function is equal to the reported confidence $c$.

Metacognitive noise is defined by a probability distribution and a metacognitive noise parameter $\sigma_m$. The appropriate noise distribution for such readout noise is an open empirical question. Here, I introduce a family of potential candidates. A key consideration for the choice of a noise distribution is the issue of sign flips. I distinguish two cases.

A first scenario is that the metacognitive level initially deals with signed decision values, such that metacognitive noise can cause sign flips of these decision values. For instance, while an observer may have issued a type 1 response for stimulus category $S^+$, readout noise could flip the sign of the decision value toward $S^-$ at the metacognitive level. How would an observer indicate their confidence in such a case? Unless confidence rating scales include the possibility to indicate errors (which I do not consider here), the only sensible response would be to indicate a confidence of 0, since confidence ratings apply to the choice made and not to the choice one would have hypothetically made based on a subsequent metacognitive representation.

Enforcing a lower bound of 0 is a form of post-hoc censoring which leads to the concept of a *censored* (or *rectified*) distribution. If a distribution is left-censored at zero, all negative parts of the

distribution are assigned to the probability mass of zero, resulting in a distribution with a discrete term at $z = 0$ and a continuous term for $z > 0$ (*Figure 5A*). In case of a normal distribution, the probability of z being exactly zero is equal to the cumulative density of the normal distribution at zero. An alternative to the normal distribution is a double exponential distribution, which allows for tail asymmetry. In particular, I here consider the Gumbel distribution which has a pronounced right tail, a property that fits recent observations regarding the skewed nature of metacognitive noise (*Shekhar and Rahnev, 2021*; *Xue et al., 2021*). Mathematical definitions of all distributions are listed in *Appendix 2—table 1*.

The second scenario is that the nature of metacognitive readout noise itself makes sign flips impossible, sparing the necessity of censoring. This required noise distributions that are bounded at zero, either naturally or by means of truncation. I first consider truncated distributions, in particular the truncated normal and the truncated Gumbel distribution (*Figure 5B*). Truncating a distribution means to cut off the parts of the distribution outside the truncation points (here the range below zero) and to renormalize the remainder of the distribution to 1.

While truncated distributions behave well mathematically, compared to censored distributions it is much less clear how a natural process could lead to a truncated metacognitive noise distribution. Truncated distributions occur when values outside of the bounds are discarded, which clearly does not apply to confidence ratings. I thus consider truncated distributions as an auxiliary construct at this point that may nevertheless qualify as an approximation to an unknown natural process.

Finally, there are many candidates of probability distributions that are naturally bounded at zero, perhaps the most prominent one being the lognormal distribution. In addition, I consider the Gamma distribution (*Figure 5C*), which has a more pronounced lower tail and is also the connatural counterpart to the Beta distribution for noisy-report models (see next section).

## Metacognitive noise: noisy-report models

In contrast to noisy-readout models, a noisy-report model assumes that the readout noise of decision values is negligible ($z = z^*$) and that the dominant source of metacognitive noise occurs at the reporting stage: $c \sim f_m(c)$. Reporting noise itself may comprise various different sources of noise, occurring for example during the mental translation to an experimental confidence scale or in the form of visuomotor noise (e.g. when using a mouse cursor to indicate a continuous confidence rating).

A hard constraint for reporting noise is the fact that confidence scales are typically bounded between a minimum and a maximum confidence rating (reflecting the bounds [0; 1] for c in the present model). Reported confidence cannot be outside these bounds, regardless of the magnitude of reporting noise. As in the case of the noisy-readout model, one may consider either censored (*Figure 5D*), truncated (*Figure 5E*) or naturally bounded distributions (Beta distribution; *Figure 5F*) to accommodate this constraint.

## Metacognitive noise as a measure of metacognitive ability

As outlined above, I assume that metacognitive noise can be described either as variability during readout or report. In both cases, metacognitive noise is governed by the parameter $\sigma_m$. Higher values of $\sigma_m$ will lead to a flatter relationship between reported confidence and sensory evidence, that is, confidence ratings become more indifferent with regard to different levels of evidence (*Figure 3B*).

The behavior of the metacognitive noise parameter is closely related to the concept of metacognitive efficiency (*Fleming and Lau, 2014*), a term coined for measures of metacognitive ability that aim at being invariant to type 1 performance (in particular, $M_{ratio}$). As outlined in the introduction, the type 1 performance independence of $M_{ratio}$ has been contested to some degree, on the basis of empirical data and as well as in simulations that consider the presence of metacognitive noise (*Bang et al., 2019*; *Guggenmos, 2021*).

Here, I was interested in two main questions: can metacognitive noise $\sigma_m$ be truthfully recovered regardless of type 1 performance? And further, to what degree are metacognitive noise $\sigma_m$ and metacognitive efficiency correlated and thus potentially capture similar constructs?

To assess the type 1 performance dependency, I simulated data with varying levels of sensory noise $\sigma_s$ and five different values of $\sigma_m$. In each case I computed $M_{ratio}$ on the data and also fitted the model to recover the metacognitive noise parameter $\sigma_m$. As shown in the left panels of *Figure 6A* (noisy-report) and 6B (noisy-readout), $M_{ratio}$ shows a nonlinear dependency with varying type 1 performance levels.

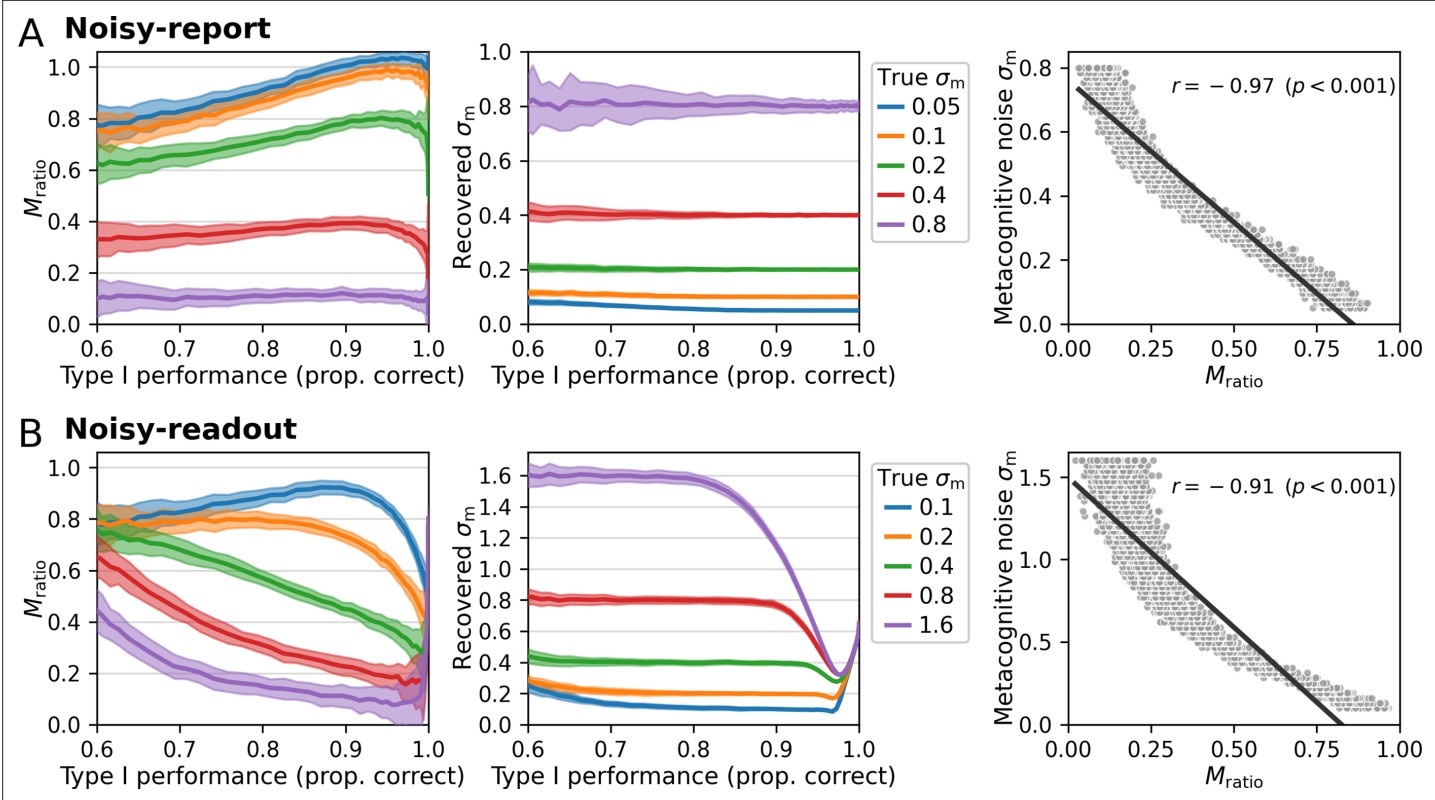

**Figure 6.** Comparison of $M_{ratio}$ and metacognitive noise $\sigma_m$. Different performance levels were induced by varying the sensory noise of the forward model. Five different levels of metacognitive noise were simulated for a truncated normal noise distribution, covering the range between low and high metacognitive noise. While $M_{ratio}$ showed a nonlinear dependency with varying type 1 performance levels both for (**A**) noisy-report models and (**B**) noisy-readout models, the recovered metacognitive noise parameter $\sigma_m$ was largely independent of type 1 performance. Shaded areas indicate standard deviations across 100 simulated subjects. Right panels: Relationship between metacognitive noise and $M_{ratio}$. Simulated data were generated with a range of varying metacognitive noise parameters $\sigma_m$ and constant sensory noise ($\sigma_s = 0.5$; proportion correct responses: 0.82). Computed $M_{ratio}$ values show a clear negative correspondence with $\sigma_m$, reflecting the fact that metacognitive performance decreases with higher metacognitive noise.

The online version of this article includes the following figure supplement(s) for figure 6:

**Figure supplement 1.** Comparison of $M_{ratio}$ and metacognitive noise $\sigma_m$ for constant stimuli.

**Figure supplement 2.** Type 1 dependency of $M_{ratio}$ and metacognitive noise $\sigma_m$ for various settings of other parameters.

While this simulation was based on multiple stimulus levels, a similar nonlinear dependency is also present for a scenario with constant stimuli (*Figure 6—figure supplement 1*).

By contrast, the parameter $\sigma_m$ is recovered without bias across a broad range of type 1 performance levels and at different levels of generative metacognitive noise (*Figure 6*, middle panels). The exception is a regime with very high metacognitive noise and low sensory noise under the noisy-readout model, in which recovery becomes biased. A likely reason is related to the inversion of the link function, which is necessary for parameter inference in noisy-readout models ('Metacognitive level'): since the link function is dependent on sensory noise $\sigma_s$, its inversion becomes increasingly imprecise as $\sigma_s$ approaches very small or very high values. However, apart from these extremal cases under the noisy-readout model, $\sigma_m$ is largely unbiased and is thus a promising candidate to measure metacognitive ability independent of type 1 performance. *Figure 6—figure supplement 2* shows that this conclusion also holds for various settings of other model parameters.

Despite the fact that $M_{ratio}$ may not be entirely independent of type 1 performance, it is likely that it captures the metacognitive ability of observers *to some degree*. It is thus interesting to assess the relationship between the model-based measure of metacognitive noise $\sigma_m$ and $M_{ratio}$. To this aim, I performed a second simulation in which type 1 performance was kept constant (at around 82% correct) by using a fixed sensory noise parameter ($\sigma_s = 0.5$) while varying the generative metacognitive noise parameter $\sigma_m$. In addition, $M_{ratio}$ was computed for each simulated observer. As shown in

the right panels of *Figure 6A and B*, there was indeed a strong negative correlation between $\sigma_m$ and $M_{ratio}$ both for the noisy-report (r = −0.97) and the noisy-readout model (r = −0.91). Of note, a very similar relationship is observed for the unnormalized measure *meta-d'* (noisy-report: r = −0.97; noisy-readout: r = −0.91). The negative sign of the correlation is expected since a higher degree of noise should lead to more imprecise confidence ratings and thus reduced metacognitive performance.

## Model fitting

Model fitting proceeds in a two-stage process. First, parameters of the sensory level are fitted by maximizing the likelihood of the model with respect to the observed type 1 decisions. Second, using the decision values predicted by the sensory level, the parameters of the metacognitive level are fitted by maximizing the likelihood with respect to observed confidence reports. The two levels are thus fitted independently. The reason for the separation of both levels is that choice-based parameter fitting for psychometric curves at the type 1/sensory level is much more established and robust compared to the metacognitive level for which there are more unknowns (e.g. the type of link function or metacognitive noise distribution). Hence, the current model deliberately precludes the possibility that the estimates of sensory parameters are influenced by confidence ratings.

In the following, the capital letter $D$ denotes observed type 1 decisions, and the capital letter $C$ denotes observed confidence ratings. The set of parameters of the sensory level is denoted as $\mathcal{P}_s := \{\sigma_s, \vartheta_s, \delta_s\}$ and the set of parameters of the metacognitive level as $\mathcal{P}_m := \{\sigma_m, \varphi_m, \delta_m, \lambda_m, \kappa_m\}$.

### Sensory level

At the sensory level, sensory noise is considered to follow a logistic distribution (*Equation 2*). The likelihood $\mathcal{L}$ of a particular type 1 decision D for stimulus x has an analytic solution given by the logistic function:

$$\mathcal{L}\left(D = S^+ \mid \mathcal{P}_s\right) = 1 - \mathcal{L}\left(D = S^- \mid \mathcal{P}_s\right) = \frac{1}{1 + \exp\left(-\frac{\pi}{\sqrt{3}\sigma_s} y^*\left(x; \vartheta_s, \delta_s\right)\right)} \tag{7}$$

where $y^*(x; \vartheta_s, \delta_s)$ is given by *Equation 1*. By maximizing the (cumulative) likelihood across trials, estimates for $\sigma_s, \vartheta_s$, and $\delta_s$ are obtained.

### Metacognitive level

Parameter inference at the metacognitive level requires the output of the sensory level (decision values $y$) and empirical confidence ratings $C$. In addition, if the goal is to compute confidence as probability correct (as assumed here), the estimate of sensory noise $\sigma_s$ is required. By running the model in feed-forward mode and using the fitted sensory parameters, the likelihood of confidence ratings is evaluated either at the stage of readout (noisy-readout model) or report (noisy-report model).

Special consideration is necessary for the noisy-readout model in which the significant metacognitive noise source is assumed at the level of an unobserved variable – metacognitive evidence. For this reason, the model must be inverted from the point of the observed variable (here confidence ratings) into the space of the latent variable (metacognitive evidence). A consequence of this is that the link function that transforms metacognitive decision values to confidence ratings must be strictly monotonically increasing in the noisy-readout scenario, as model inversion would otherwise be ambiguous.

Using the link function considered for this work, the tangens hyperbolicus (*Equation 5*), the inversion is as follows:

$$Z = \frac{2\sqrt{3}\sigma_s}{\pi} \operatorname{arctanh}\left(\frac{C - \kappa_m}{\lambda_m}\right) \tag{8}$$

Importantly, the likelihood $\mathcal{L}\left(C \mid \mathcal{P}_m\right)$ of observed confidence ratings $C$ given parameters $\mathcal{P}_m$ not only depends on the uncertainty of the model prediction for metacognitive decision values $z^*(y)$, but also on the uncertainty around the decision values $y$ themselves. Computing the likelihood $\mathcal{L}\left(C \mid \mathcal{P}_m\right)$ thus requires an integration over the probability density $f_s(y)$:

$$\text{Noisy-readout}: \quad \mathcal{L}\left(C \mid \mathcal{P}_m\right) = \int f_m\left(Z \mid z^*(y)\right) f_s(y)\ dy \tag{9}$$

The term $z^*(y)$ is given by *Equation 4*.

In case of the noisy-report model, the likelihood can be directly computed with respect to the observed confidence reports $C$, that is, without inversion of the link function:

$$\text{Noisy-report:} \qquad \mathcal{L}\left(C \mid \mathcal{P}_{\mathrm{m}}\right) = \int f_{\mathrm{m}}\left(C \mid c^{*}\left(y\right)\right) f_{\mathrm{s}}\left(y\right) \ \mathrm{d}y \tag{10}$$

The term $c^*(y)$ corresponds to the link function in *Equation 6*.

## Parameter recovery

To ensure that the model fitting procedure works as expected and that model parameters are distinguishable, I performed a parameter recovery analysis. To this end, I systematically varied each parameter of a model with metacognitive evidence biases and generated data (see below, for a model with confidence biases). Specifically, each of the six parameters ($\sigma_{\mathrm{s}}, \vartheta_{\mathrm{s}}, \delta_{\mathrm{s}}, \sigma_{\mathrm{m}}, \varphi_{\mathrm{m}}, \delta_{\mathrm{m}}$) was varied in 500 equidistant steps between a sensible lower and upper bound. The model was then fit to each dataset to obtain the recovered parameters.

To assess the relationship between fitted and generative parameters, I computed linear slopes between each generative parameter (as the independent variable) and each fitted parameter (as the dependent variable), resulting in a 6 × 6 slope matrix. Slopes instead of correlation coefficients were computed, as correlation coefficients are sample-size-dependent and approach 1 with increasing sample size even for tiny linear dependencies. Thus, as opposed to correlation coefficients, slopes quantify the strength of a relationship. To reduce the sensitivity to outliers, slopes were computed using the Theil-Sen method which is based on the median of the slopes of all lines through pairs of points (*Sen, 1968*; *Theil, 1950*). Comparability between the slopes of different parameters is given because (i) slopes are – like correlation coefficients – expected to be 1 if the fitted values precisely recover the true parameter values (i.e. the diagonal of the matrix) and (ii) all parameters have a similar value range which allows for a comparison of off-diagonal slopes at least to some degree.

To test whether parameter recovery was robust against different settings of the respective other parameters, I performed this analysis for a coarse parameter grid consisting of three different values for each of the six parameters except $\sigma_{\mathrm{m}}$, for which five different values were considered. This resulted in $3^5 \cdot 5^1 = 1{,}215$ slope matrices for the entire parameter grid.

*Figure 7* shows the result of this analysis both for a noisy-report and a noisy-readout model, expanded along the sensory ($\sigma_{\mathrm{s}}$) and metacognitive ($\sigma_{\mathrm{m}}$) noise axis of the coarse parameter grid. Overall, generative and fitted parameters show excellent correspondence, that is, nearly all slopes on the diagonal are close to 1.

Off-diagonal slopes indicate a potential trade-off between different parameters in the fitting procedure. In the present analysis, the only marked trade-off emerges between metacognitive noise $\sigma_{\mathrm{m}}$ and the metacognitive evidence biases ($\varphi_{\mathrm{m}}, \delta_{\mathrm{m}}$) in the noisy-readout model, under conditions of low sensory noise. In this regime, the multiplicative evidence bias $\varphi_{\mathrm{m}}$ becomes increasingly underestimated and the additive evidence bias $\delta_{\mathrm{m}}$ overestimated with increasing metacognitive noise. Closer inspection shows that this dependency emerges only when metacognitive noise is high – up to $\sigma_{\mathrm{m}} \approx$ 0.3 no such dependency exists. It is thus a scenario in which there is little true variance in confidence ratings (due to low sensory noise many confidence ratings would be close to 1 in the absence of metacognitive noise), but a lot of measured variance due to high metacognitive noise. It is likely for this reason that parameter inference is problematic. Overall, except for this arguably rare scenario, all parameters of the model are highly identifiable and separable.

While this analysis was carried out for 500 trials per simulated subject to assess the scenario of a typical metacognition study, *Figure 7—figure supplement 1* shows the same analysis with 10,000 trials to give an indication of the theoretical linear dependency structure.

I repeated the same analysis for a model with metacognitive confidence biases. The result of this analysis shows that also the parameters of a model with metacognitive confidence biases can be accurately recovered (*Figure 7—figure supplement 2*). In addition, I assessed models that feature a mix of metacognitive evidence and confidence biases (*Figure 7—figure supplement 3*). The results of these analyses indicate that models with up to three bias parameters show generally good parameter recovery. An exception are models with both confidence bias parameters ($\lambda_{\mathrm{m}}, \kappa_{\mathrm{m}}$) which additionally consider one of the evidence bias parameters ($\varphi_{\mathrm{m}}$ or $\delta_{\mathrm{m}}$). For these models, considerable trade-offs

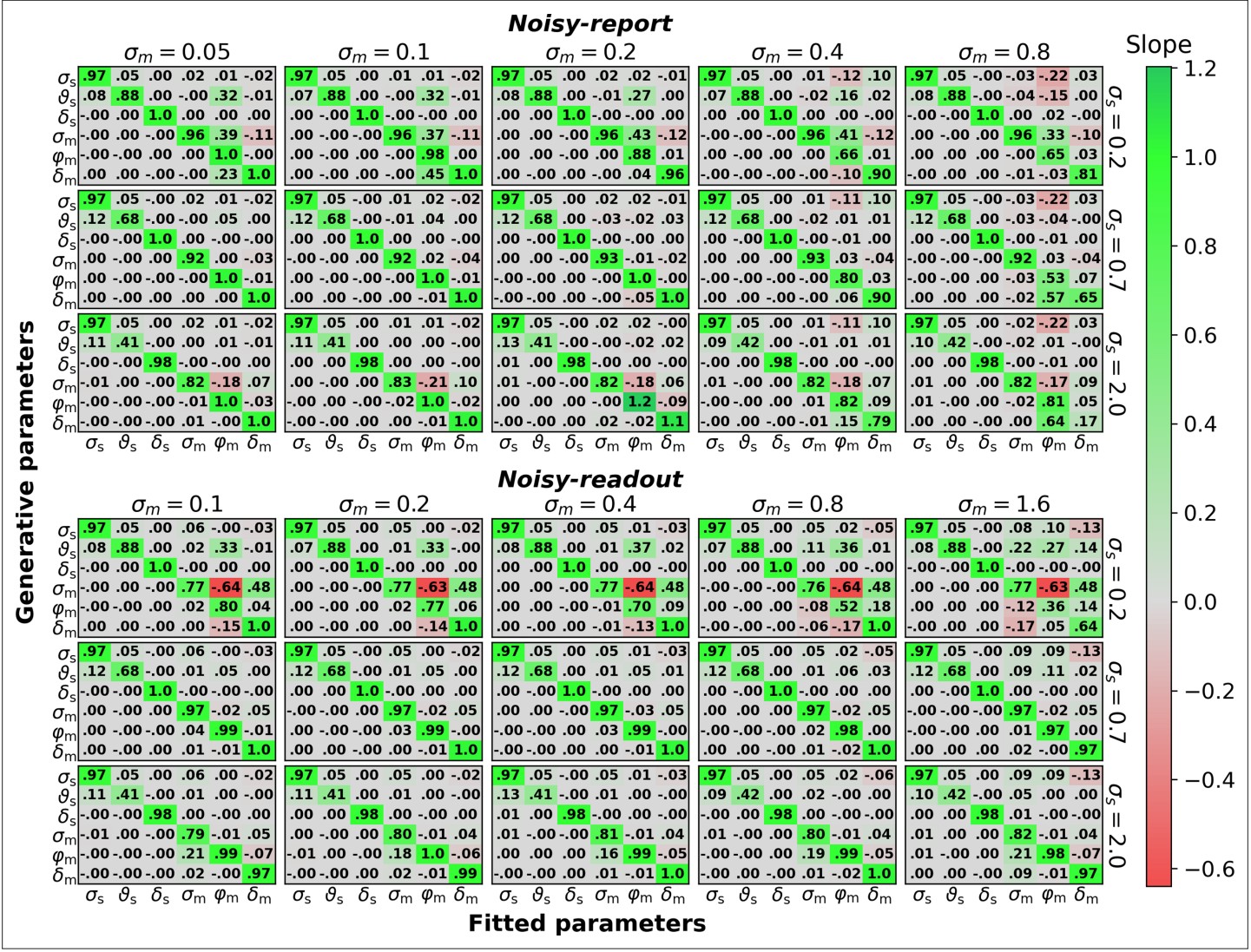

**Figure 7.** Parameter recovery (500 trials per observer). Linear dependency between generative parameters and fitted parameters for the six parameters of the noisy-report and noisy-readout model ($\sigma_s$, $\vartheta_s$, $\delta_s$, $\sigma_m$, $\varphi_m$, $\delta_m$). Linear dependency between generative and fitted parameters was assessed through robust linear slopes. The optimal value for diagonal elements is 1 while off-diagonal elements should be close to zero. Multiple slope matrices were computed for each node of a coarse parameter grid (see text). The figure thus shows average slope matrices, expanded along the coarse parameter grid axes for sensory noise $\sigma_s$ and metacognitive noise $\sigma_m$. The row-wise values for $\sigma_s$ and the column-wise values for $\sigma_m$ indicate the parameter values used for data generation, except when $\sigma_s$ or $\sigma_m$ where themselves varied.

The online version of this article includes the following figure supplement(s) for figure 7:

**Figure supplement 1.** The figure mirrors the parameter recovery analysis in *Figure 7* with 10,000 instead of 500 trials.

**Figure supplement 2.** The figure mirrors the parameter recovery analysis in *Figure 7* for a model with metacognitive confidence biases ($\lambda_m$, $\kappa_m$) instead of metacognitive evidence biases and for either 500 or 10,000 trials.

**Figure supplement 3.** Parameter recovery for a mix of evidence-related and confidence-related metacognitive bias parameters.

**Figure supplement 4.** No indication of biases in parameter recovery.

**Figure supplement 5.** Parameter recovery across a range of trial numbers (500 to 10,000).

**Figure supplement 6.** Model recovery.

between the bias parameters start to emerge. Finally, a model with all four considered metacognitive bias largely fails to recover its bias parameters.

While the previous analysis indicates overall excellent parameter recovery performance, there nevertheless could be certain biases in parameter recovery that escape a slope-based analysis. To test

for such biases, in *Figure 7—figure supplement 4* I assessed the precise values of recovered parameters across a range of generative parameter values. In all instances, the model precisely recovered the input parameter values, thereby demonstrating the absence of systematic biases.

Finally, to more systematically assess the precision of parameter recovery in dependence of trial number, I set the value of each generative parameter to 0.2 (arbitrary value) and tested parameter recovery across a range of trial numbers between 500 and 10,000. The results in *Figure 7—figure supplement 5* provide a reference for the expected precision of parameter estimates in dependence of trial number.

## Model recovery

One strength of the present modeling framework is that it allows testing whether inefficiencies of metacognitive reports are better described by metacognitive noise at readout (noisy-readout model) or at report (noisy-report model). To validate this type of application, I performed an additional model recovery analysis which tested whether data simulated by either model are also best fitted by the respective model.

*Figure 7—figure supplement 6* shows that the recovery probability was close to 1 in most cases, thus demonstrating excellent model identifiability. With fewer trials per observer, recovery probabilities decreased expectedly, but were still at a reasonable level. The only edge case with poorer recovery was a scenario with low metacognitive noise and high sensory noise. Model identification is particularly hard in this regime because low metacognitive noise reduces the relevance of the metacognitive noise source, while high sensory noise increases the general randomness of responses.

## Application to empirical data

### On using the model framework

The present work does not propose a single specific model of metacognition, but rather provides a flexible framework of possible models and a toolbox to engage in a metacognitive modeling project. Applying the framework to an empirical dataset thus requires a number of user decisions: which metacognitive noise type is likely more dominant? which metacognitive biases should be considered? which link function should be used? These decisions may be guided either by a priori hypotheses of the researcher or can be informed by running a set of candidate models through a statistical model comparison.

As an exemplary workflow, consider a researcher who is interested in quantifying overconfidence in a confidence dataset with a single parameter to perform a brain-behavior correlation analysis. The concept of under/overconfidence already entails the first modeling choice, as only a link function that quantifies probability correct (*Equation 6*), i.e. statistical confidence, allows for a meaningful interpretation of metacognitive bias parameters. Moreover, the researcher must decide for a specific metacognitive bias parameter. The researcher may not be interested in biases at the level of the confidence report, but, due to a specific hypothesis, rather at metacognitive biases at the level of readout/evidence, thus leaving a decision between the multiplicative and the additive evidence bias parameter. Also, the researcher may have no idea whether the dominant source of metacognitive noise is at the level of the readout or report. To decide between these options, the researcher computes the evidence (e.g., AIC) for all four combinations and chooses the best-fitting model (ideally, this would be in a dataset independent from the main dataset).

### Application to an example dataset (Shekhar and Rahnev, 2021)

To test the proposed model on real-world empirical data, I used a data set recently published by *Shekhar and Rahnev, 2021* which has a number of advantageous properties for a modeling approach. First, a high number of 2,800 trials were measured for each of the 20 participants, enabling a precise estimate of computational parameters (*Figure 7—figure supplement 5*). Second, the task design comprised multiple stimulus intensities, which is expected to improve the fit of a process model. And third, participants rated their confidence on a continuous scale. While the model works well with discrete confidence ratings, only continuous confidence scales harness the full expressive power of the model. In each trial, participants indicated whether a Gabor patch imposed on a noisy background was tilted counterclockwise or clockwise from a vertical reference and simultaneously rated their confidence. The average performance was 77.7% correct responses.

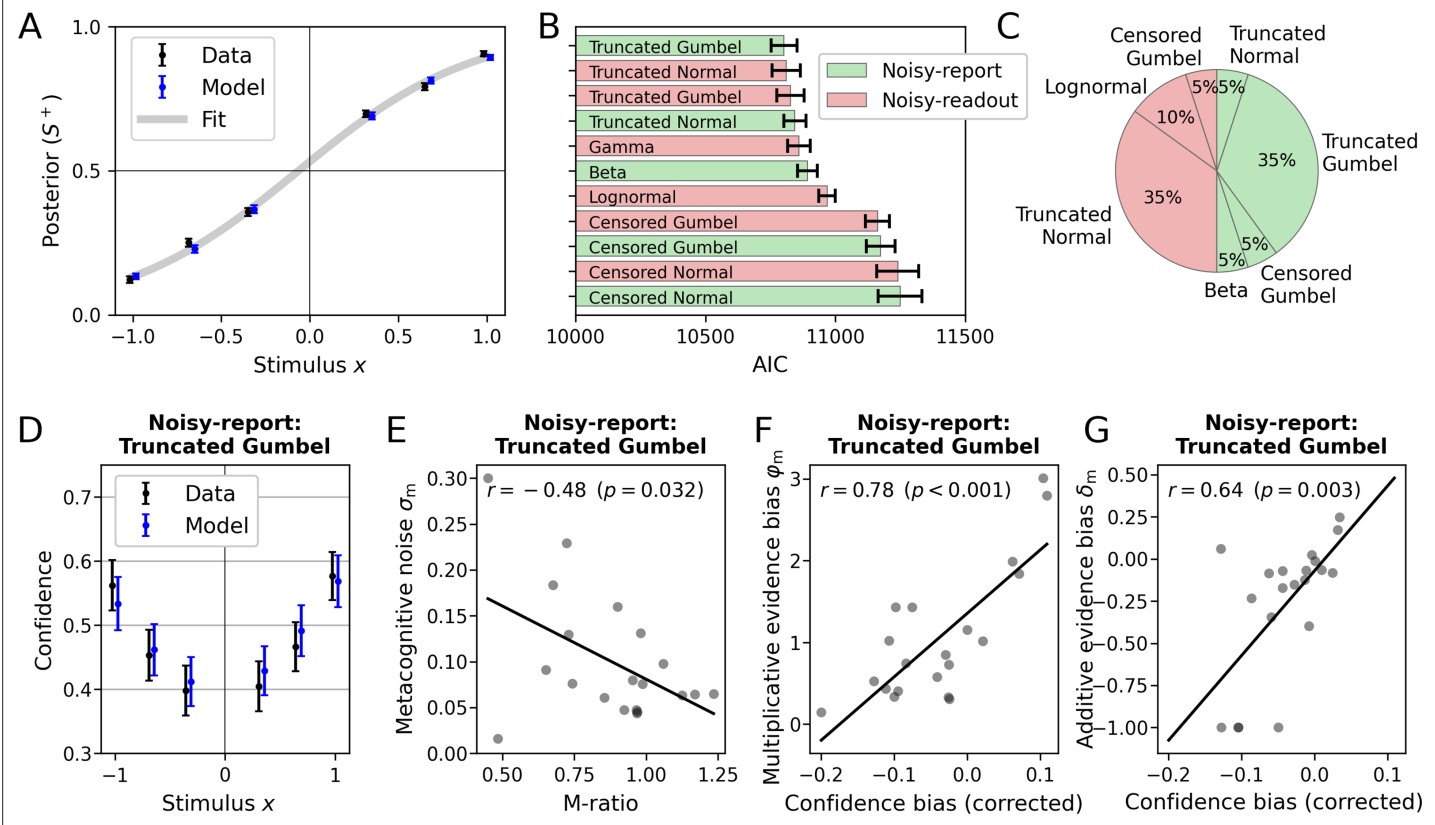

**Figure 8.** Application of the model to empirical data from *Shekhar and Rahnev, 2021* (N=20). (**A**) Posterior probability (choice probability for $S^+$) as a function of normalized signed stimulus intensity. Model-based predictions closely follow the empirical data. Means and standard errors across subjects were computed for the three difficulty levels of each stimulus category. The fit is based on a logistic function with a sensory bias parameter $\delta_s$. (**B**) Comparison of noisy-readout and noisy-report models featuring different metacognitive noise distributions. Model comparison was based on the Akaike information criterion (AIC) which quantified model evidence at the metacognitive level (the sensory level is identical between models). Error bars indicate standard errors of the mean (SEM). (**C**) Breakdown of best-fitting models across participants. (**D–G**) Inspection of the metacognitive level for the winning model of the type *noisy-report* with a truncated Gumbel noise distribution. (**D**) Empirical confidence is well-fitted by model-based predictions of confidence which are based on an average of 1000 runs of the generative model. Error bars represent SEM. (**E**) Relationship of empirical $M_{ratio}$ and model-based metacognitive noise $\sigma_m$. (**F**) Partial correlation of the empirical confidence bias the and model-based multiplicative evidence bias $\varphi_m$. The additive evidence bias was partialed out from the confidence bias. (**G**) Partial correlation of the empirical confidence bias and the model-based additive evidence bias $\delta_m$. The multiplicative evidence bias was partialed out from the confidence bias.

The online version of this article includes the following figure supplement(s) for figure 8:

**Figure supplement 1.** Empirical confidence distributions and generative models of all 20 subjects in *Shekhar and Rahnev, 2021*.

*Figure 8A* visualizes the overall model fit at the sensory level. The posterior, defined as the probability of choosing $S^+$, closely matched the model fit. The average posterior probability showed a slight x-offset toward higher choice probabilities for $S^+$ which was reflected in a positive average sensory bias $\delta_s$ (group mean ± SEM = 0.06 ± 0.03). Since no stimulus intensities near chance-level performance were presented to participants, a sensory threshold parameter $\vartheta_s$ was not fitted.

At the metacognitive level, I compared noisy-readout and noisy-report models in combination with the metacognitive noise distributions introduced in Result, 'Metacognitive noise: noisy-readout models' and 'Metacognitive noise: noisy-report models'. For this analysis, I considered metacognitive *evidence* biases only (i.e. multiplicative evidence bias $\varphi_m$ and additive evidence bias $\delta_m$). The model evidence was computed based on the Akaike information criterion (AIC; *Akaike, 1974*). As shown in *Figure 8B*, with the exception of censored distributions, all models performed at a similar level. Seven of the 10 tested models were the winning model for at least one participant (*Figure 8C*).

Interestingly, there were quite clear patterns between the shapes of individual confidence distributions and the respective winning model (*Figure 8—figure supplement 1*). For instance, a single participant was best described by a noisy-report+Beta model, and indeed the confidence distribution

of this participant is quite unique and plausibly could be generated by a Beta noise distribution (participant 7). Participants who were best fitted by noisy-readout models have quite specific confidence distributions with pronounced probability masses at the extremes and very thin coverage at intermediate confidence levels (participants 4–6, 8, 10, 13, 19) – except those, for which the lognormal readout noise distribution was optimal (participants 9 and 11). Finally, two participants were best fitted by a censored distribution (participants 14 and 16), contrary to the general tendency. These participants likewise had fairly idiosyncratic confidence distributions characterized by the combination of a probability mass centered at mid-level confidence ratings and a prominent probability mass at a confidence of 1. While a more detailed analysis of individual differences is beyond the scope of this paper, these examples may point to distinct phenotypes of metacognitive noise.

In the next step, I inspected the winning metacognitive model (noisy report +truncated Gumbel) in more detail. While the selection of this specific model is arbitrary due to the similar performance of several other models, it serves the illustrative purpose and the differences between these models were overall negligible.

I first compared confidence ratings predicted by the model with empirical confidence ratings across the range of experimental stimulus intensities. As shown in *Figure 8D*, model-predicted confidence tracked behavioral confidence quite well (*Figure 8D*). This included a slight confidence bias toward $S^+$, which itself is likely a result of the general sensory bias toward $S^+$.

I then compared the fitted parameter values of the model with conventional behavioral measures of metacognition. In Results, 'Metacognitive noise as a measure of metacognitive ability', a tight inverse relationship between metacognitive efficiency ($M_{ratio}$) and the metacognitive noise parameter $\sigma_m$ was demonstrated for simulated data. As shown in *Figure 8E*, for the empirical data there was likewise a negative relationship, although weaker ($r_{Pearson} = -0.48$, $P = 0.032$). Note that this relationship is by no means self-evident, as $M_{ratio}$ values are based on information that is not available to a process model: *which specific* responses are correct or incorrect. I will elaborate more on this aspect in the discussion, but assert for now that metacognitive efficiency in empirical data can, at least in part, be accounted for by modeling metacognitive noise in a process model.

As outlined above, the multiplicative evidence bias $\varphi_m$ and the additive evidence bias $\delta_m$ can be interpreted as metacognitive biases. To assess the validity of these parameters, I computed individual confidence biases by subtracting the participants' objective accuracy from their subjective accuracy (based on confidence ratings). Positive and negative values of this confidence bias are often regarded as evidence for over- and underconfidence. As shown in *Figure 8F and G*, both parameters show the expected relationships: higher individual confidence biases are associated with higher values of $\delta_m$ when controlling for $\varphi_m$ ($r_{Partial} = 0.78$, p < 0.001), and with $\varphi_m$ when controlling for $\delta_m$ ($r_{Partial} = 0.64$, p = 0.003). This analysis confirms that the metacognitive bias parameters of the model meaningfully relate to the over- and underconfidence behavior in empirical data.

In a final step, I focus on the model fit of a single participant (*Figure 9*). The selected participant has a relatively high degree of sensory noise (proportion correct = 0.74; $\sigma_s = 1.04$) compared to the group mean (proportion correct ± SEM = 0.78 ± 0.01; $\sigma_s$ ± SEM = 0.89 ± 0.04), reflected in a relatively flat psychometric curve (*Figure 9A*). Like many participants in the dataset, the participant tends to disproportionally choose clockwise/$S^+$ over counterclockwise/$S^-$, reflected in a psychometric curve shifted toward $S^+$ and hence a positive response bias ($\delta_s = 0.23$).

*Figure 9B and C* visualize the results of the metacognitive level, which is again of the type noisy-report+truncated Gumbel. For this participant, the model fit indicates a negative additive evidence bias $\delta_m$, thereby introducing a threshold below which stimuli are not metacognitively accessible (indicated by a flat region for the link function in *Figure 9B*). This negative additive evidence bias is compensated by a relatively high multiplicative evidence bias $\varphi_m = 1.15$, resulting in an average confidence of 0.488 that is close to the group average (0.477 ± 0.038).

While below average in terms of type 1 performance, this participant excels in terms of metacognitive performance. This is both indicated by a high $M_{ratio}$ of 1.23 (group mean ± SEM = 0.88 ± 0.05) and a low metacognitive noise parameter $\sigma_m$ of 0.06 (group mean ± SEM = 0.10 ± 0.02).

It is important to note that a low metacognitive noise parameter $\sigma_m$ does not imply that the participants' confidence ratings are expected to be within a narrow range for each specific stimulus intensity. This is because the uncertainty of the sensory level translates to the metacognitive level: the width of decision value distributions, as determined by sensory noise $\sigma_s$, also affects the expected width of

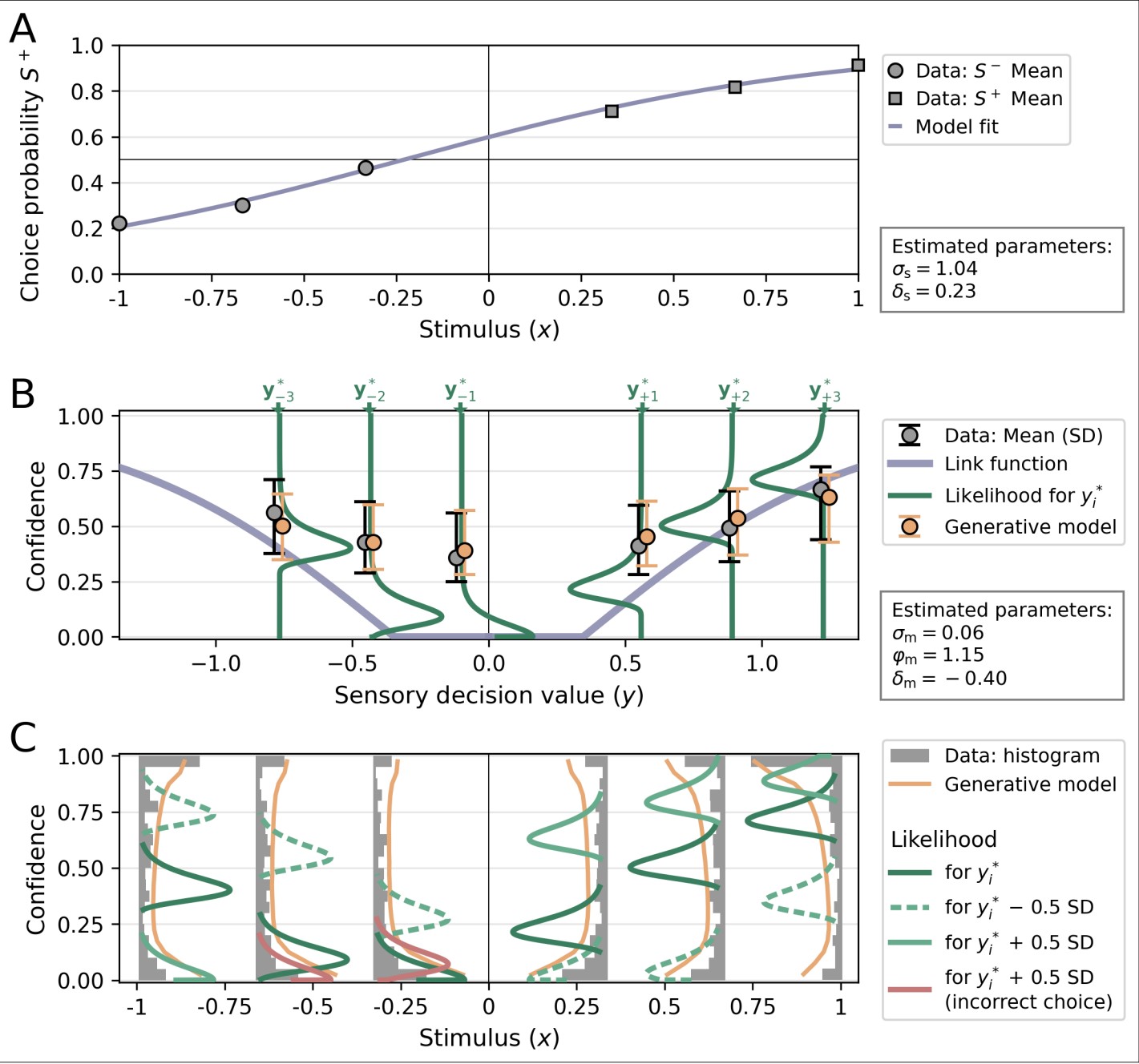

**Figure 9.** Visualization of a model fit for a single participant from *Shekhar and Rahnev, 2021*. The applied model was a noisy-report model with a metacognitive noise distribution of the type truncated Gumbel and metacognitive *evidence* biases Each stimulus category in *Shekhar and Rahnev, 2021* was presented with three intensity levels, corresponding to values of ±1/3, ±2/3, and ±1 in normalized stimulus space (variable *x*). (**A**) Choice probability for $S^+$ as a function of stimulus intensity. The positive sensory bias $\delta_s$ shifts the logistic function toward the left, thereby increasing the choice probability for $S^+$. (**B**) Link function, average confidence ratings and likelihood. The link function was transformed into decision value space *y*, for illustratory purposes. The flat range of the link function is caused by a relatively large additive evidence bias $\delta_m$. Confidence ratings from empirical data (gray) and from the generative model (orange) for each stimulus levels *i* are indicated by their mean and standard deviation. Note that these confidence averages derive from the whole range of possible decision values and they are anchored at the most likely decision values $y_i^*$ of each stimulus level *i* only for illustratory purposes. The likelihood for confidence ratings is shown only for the most likely decision values $y_i^*$ of each stimulus level *i*. (**C**) Confidence distributions and likelihood. Empirical confidence ratings are shown as a histograms and confidence ratings obtained from the generative model as line plots. To visualize the effect of sensory uncertainty on the metacognitive level, likelihood distributions are plotted not only for the most likely values $y_i^*$ of the decision value distributions, but also half a standard deviation below (dashed and lighter color) and above (solid and lighter color). The width of likelihood distributions is controlled by the metacognitive noise parameter $\sigma_m$. Distributions colored in red indicate that a sign flip of decision values has occurred, i.e. responses based on these decision values would be incorrect.

downstream confidence distributions. Indeed, the behavioral confidence distributions in *Figure 9C* are spread out across the entire confidence range for all difficulty levels. In *Figure 9C* this aspect is emphasized by not only showing the confidence likelihood for the most likely decision value $y_i^*$ of each stimulus level $i$, but also for sensory decision values 0.5 standard deviations below and above $y_i^*$.

Note that when considering decision values 0.5 standard deviations above $y_i^*$, a sign flip occurs for the two lower stimulus intensities of $S^-$ (indicated with likelihood distributions shaded in red). In these cases, the participant would make an incorrect choice. Moreover, the two lower stimulus intensities of $S^-$ show a well-known characteristic of statistical confidence: an increase of confidence for incorrect choices as stimulus difficulty increases (*Sanders et al., 2016*).

To compare the empirical confidence distribution of this participant with the distribution predicted by the model, the parameters in the generative model were set to their corresponding fitted values and sampled confidence ratings. The average predicted confidence ratings (*Figure 9B*, orange error bars) and the density histograms (*Figure 9C*, orange line plots) obtained from this sampling procedure demonstrate a close fit with the participant's confidence rating distributions. This close correspondence is not limited to this particular participant. As shown in *Figure 8—figure supplement 1*, a generative model described by $\sigma_\mathrm{m}$, $\delta_\mathrm{m}$ and $\varphi_\mathrm{m}$ is able to approximate a wide range of idiosyncratic empirical confidence distributions.

## Discussion

The present work introduces and evaluates a process model of metacognition and the accompanying toolbox *ReMeta* (see Materials and methods). The model connects key concepts in metacognition research – metacognitive readout, metacognitive biases, metacognitive noise – with the goal of providing an account of human metacognitive responses. The model can be directly applied to confidence datasets of any perceptual or non-perceptual modality.

As any cognitive computational model, the model can serve several purposes such as inference about model parameters, inference about latent variables and as a means to generate artificial data. In the present work, I focused on parameter inference, in particular metacognitive parameters describing metacognitive noise ($\sigma_\mathrm{m}$) and metacognitive biases ($\varphi_\mathrm{m}$, $\delta_\mathrm{m}$, $\lambda_\mathrm{m}$, $\kappa_\mathrm{m}$). Indeed, I would argue that this use case is one of the most pressing issues in metacognition research: parametrically characterizing the latent processes underlying human confidence reports without the confound of type 1 behavior that hampers descriptive approaches.

In the context of metacognitive biases, I have shown that the conventional method of simply comparing objective and subjective performance (via confidence ratings) is flawed not only because it is biased toward overconfidence, but also because it is strongly dependent on type 1 performance. Just as in the case of metacognitive performance, unbiased inferences about metacognitive biases thus require a process model approach.

Here, I introduced four metacognitive bias parameters loading either on metacognitive evidence or the confidence report. As shown through the simulation of calibration curves, all bias parameters can yield under- or overconfidence relative to a bias-free observer. The fact that the calibration curves and the relationships between type 1 performance and confidence biases are quite distinct between the proposed metacognitive bias parameters may indicate that these are to some degree dissociable. Moreover, in an empirical dataset the multiplicative evidence bias $\varphi_\mathrm{m}$ and the additive evidence bias $\delta_\mathrm{m}$ strongly correlated with a conventional confidence bias measure, thereby validating these parameters.

The second kind of metacognitive parameter considered in this work is metacognitive noise (*Mueller and Weidemann, 2008*; *Jang et al., 2012*; *De Martino et al., 2013*; *van den Berg et al., 2017*; *Bang et al., 2019*; *Shekhar and Rahnev, 2021*). As with metacognitive biases, metacognitive noise may arise at different stages of the processing hierarchy and in the present work I investigated two kinds: noise at readout and report. Both parameters affect the precision of confidence ratings and as a result they showed an expected negative relationship with regular measures of metacognitive ability (*meta-d'*, $M_\mathrm{ratio}$). Importantly, I show that while even $M_\mathrm{ratio}$, a measure normalized for type 1 performance, was dependent on type 1 performance for simulated data, recovered estimates of metacognitive noise were largely invariant to type 1 performance. Thus, just as in the case of metacognitive biases, the entanglement of metacognitive and type 1 behavior can be unraveled by means of a process model approach.

While this summary so far emphasized the advantages of a process model approach to metacognition, there are a number of remaining challenges. First, it is entirely possible that a comprehensive model of metacognition is non-invertible from the point of confidence ratings. This challenge is exemplified by the noisy-readout model, for which the inversion requires a strictly monotonically increasing link function. To achieve unbiased parameter inferences, one would need additional observed measures along the processing hierarchy. For instance, reaction time could be considered an implicit proxy for confidence, which is affected by readout noise but not by reporting noise. Conditional on finding an appropriate functional relationship to metacognitive evidence, reaction times could allow for an unbiased inference of metacognitive readout noise or metacognitive evidence bias parameters.

Second, the effects of different sources of bias and noise along the processing hierarchy may be so strongly correlated that their dissociation would require unrealistic amounts of confidence data. This dissociation, however, is essential for many research questions in metacognition – whether the goal is to derive a fundamental model of human metacognition or whether one is interested in specific aberrancies in mental illness. An example for the latter is the frequent observation of overconfidence in schizophrenia which is thought to reflect a more general deficit in the ability to integrate disconfirmatory evidence (*Speechley et al., 2010*; *Zawadzki et al., 2012*) and may underlie the maintenance of delusional beliefs (*Moritz and Woodward, 2006b*). To investigate this specific hypothesis, it is central to dissociate whether metacognitive biases mainly apply at the reporting stage – which may be a result of the disease – or at an earlier metacognitive processing stage, which may be involved in the development of the disease. This issue likewise could be addressed by measuring behavioral, physiological or neurobiological processes that precede the report of confidence.

Third, the demonstration of an unbiased recovery of metacognitive noise and bias parameters in a process model approach comes with a strong caveat, since the data is generated with the very same model that is used for parameter recovery. Yet, all models are wrong, starts a famous saying, and this certainly applies to current models of metacognition. The question is thus: given the unknown true model that underlies empirical confidence ratings, to what degree can parameters obtained from an approximated model be considered unbiased? The way forward here is to continuously improve computational models of metacognition in terms of model evidence, thus increasing the chances that fitted parameters are meaningful estimates of the true parameters.

With respect to previous modeling work, a recent paper by *Shekhar and Rahnev, 2021* deserves special attention. Here too, the authors adopted a process model approach for metacognition with the specific goal of deriving a measure of metacognitive ability, quite similar to the metacognitive noise parameter $\sigma_m$ in this work. One key difference is that Shekhar and Rahnev tailored their model to discrete confidence scales, such that each possible confidence rating (for each choice option) is associated with a separately fitted confidence criterion (as notable precursor of this idea is *Adler and Ma, 2018a*). This introduces maximal flexibility, as essentially arbitrary mappings from internal evidence to confidence can be fitted. In addition, it requires minimal assumptions about the link functions that underlies the computation of confidence, apart from an ordering constraint applied to the criteria.

However, while this flexibility is a strength, it also comes at certain costs. One issue is the relatively large number of parameters that have to be fitted. Shekhar and Rahnev note that the MLE procedures for the fitting of confidence criteria often got stuck in local minima. Rather than via MLE, confidence criteria were thus fitted by matching the expected proportion of high confidence trials to the observed proportion for each criterion. It is thus not guaranteed that the obtained confidence criterions indeed maximize the likelihood under the data. Furthermore, to make a criterion-based model compatible with data from a continuous confidence scale, confidence reports have to be discretized. Apart from the loss of information associated with discretization, this introduces uncertainty as to how exactly the data should be binned (e.g. equinumerous versus equidistant). Another aspect worth mentioning is that a criterion-based approach effectively corresponds to a stepwise link function, which is not invertible. Making inferences about readout noise thus poses a challenge to such criterion-based models.

In the present work, I assumed a mapping between internal evidence and confidence that can be described by a parametric link function. This too comes with advantages and disadvantages. On the one hand, a parametric link function naturally imposes strong constraints on the mapping between internal evidence and confidence. In reality, this mapping might not conform to any simple function – and even if it did, different observers might apply different functions. On the other hand, imposing a specific link function can be seen as a form of regularization when statistical power is insufficient to

constrain a large number of individual criteria. Further, a parametric link function does not need to worry about the discretization of confidence ratings, while still being compatible with a priori discretized ratings. Finally, a meaningful inference about metacognitive biases requires a parametric link function which computes the subjective probability of being correct (as in *Equation 5*).

The process model approach deviates in an important way from standard analyses of confidence reports based on the type 2 receiver operating curve. As type 2 ROC analyses are solely based on stimulus-specific type 1 and type 2 responses, they do not consider one of the arguably most important factors in this context: stimulus intensity. This implies that such measures cannot dissociate to what degree variability in confidence ratings is based on stimulus variability or on internal noise. In contrast, since a process model specifies the exact transformation from stimulus intensity to decision variable to confidence, this source of variance is appropriately taken into account. The metacognitive noise parameter $\sigma_m$ introduced here is thus a measure of the *unexpected* variability of confidence ratings, after accounting for the variability on the stimulus side. Note that such stimulus variability is typically present even in designs with intended constant stimulus difficulty, due to the involvement of randomness in the generation of unique trial-by-trial stimuli. In many cases, the *effective* stimulus difficulty (i.e. including this random component of stimulus variability) can likewise be quantified using appropriate feature-based energy detectors (see e.g. *Guggenmos et al., 2016*).

The process model approach bears another important difference compared with type 2 ROC analyses, in this case a limiting factor on the side of the process model. As the area under the type 2 ROC quantifies to what degree confidence ratings discriminate between correct and incorrect responses, it is important to recognize what valuable piece of information the correctness of a *specific* response is. Over and above stimulus intensity, the correctness of a response will typically be influenced by negative factors such as attentional lapses, finger errors, tiredness, and positive factors such as phases of increased motivation or concentration. All of these factors not only influence type 1 performance, but they also influence the type 2 response that one would expect from an ideal metacognitive observer. Analyses of type 2 ROCs implicitly make use of this information insofar as they consider the correctness of each individual response.

In contrast, the information about the objective trial-by-trial accuracy is not available in a process model. The signal that enters the metacognitive level of the process model is based only on information that was accessible to the observer (in particular, sensory decision variables), but not based on the correctness of specific choices, which is only accessible to the experimenter. Note that this is not a limitation specific to the present model, but the nature of process models in general. Improving process models in this regard requires additional measurements that reflect knowledge of the observer, such as subjective reports of attentional lapses or finger errors.

In sum, while a type 2 ROC analysis – as a descriptive approach – does not allow any conclusions about the causes of metacognitive inefficiency, it is able to capture a more thorough picture of metacognitive sensitivity: that is, it quantifies metacognitive awareness not only about one's own sensory noise, but also about other potential sources of error (attentional lapses, finger errors, etc.). While it cannot distinguish between these sources, it captures them all. On the other hand, only a process model approach will allow to draw specific conclusions about mechanisms – and pin down sources – of metacognitive inefficiency, which arguably is of major importance in many applications.

Finally, how does the present model relate to the recent discussion between Bayesian and Non-Bayesian models of confidence (*Aitchison et al., 2015*; *Sanders et al., 2016*; *Adler and Ma, 2018a*)? A Bayesian observer of the (inner) world is one who maintains a posterior probability density over possible states of that world. In particular, computing confidence for such an observer corresponds to integrating the posterior over all possible states for which the type 1 choice would be correct. In this sense, the model proposed here with the link function provided in *Equation 5* corresponds to a Bayesian observer, albeit one that can be susceptible to metacognitive biases and to additional sources of metacognitive noise. Thus, while the observer is Bayesian in nature, it may not be Bayes optimal. At the same time, the framework and the toolbox are flexible to allow for 'non-Bayesian' link functions (*Figure 3—figure supplement 1*) that could represent certain idiosyncratic heuristics and shortcuts inherent to human confidence judgements. Of note, the model proposed here does not consider prior distributions over the stimulus categories (see e.g. *Adler and Ma, 2018a*). Instead, it is assumed that the observer considers both stimulus categories equally likely which is considered a reasonable assumption if stimulus categories are balanced.

## Conclusion

The model outlined in this paper casts confidence as a noisy and potentially biased transformation of sensory decision values. The model parameters that shape this transformation provide a rich account of human metacognitive inefficiencies and metacognitive biases. In particular, I hope that the underlying framework will allow a systematic model comparison in future confidence datasets to elucidate sources of metacognitive noise, to narrow down candidate noise distributions and to differentiate between different kinds of metacognitive biases. The accompanying toolbox *ReMeta* provides a platform for such investigations.

## Materials and methods

### The *ReMeta* toolbox

The code underlying this work has been bundled in a user-friendly Python toolbox (*ReMeta*) which is published alongside this paper at https://github.com/m-guggenmos/remeta, (copy archived at swh:1:rev:43ccbf2e35b1e934dab83e156e4fbb22ac160cd2; *Guggenmos, 2022*). While its core is identical to the framework outlined here, it offers a variety of additional parameters and settings. In particular, it allows fitting separate values for each parameter depending on the sign of the stimulus (for sensory parameters) or the decision value (for metacognitive parameters). Moreover, it offers various choices for noise distributions and link functions, including criterion-based link functions.

The *ReMeta* toolbox has a simplified interface such that in the most basic case it requires only three 1-d arrays as input: stimuli, choices and confidence. The output is a structure containing the fitted parameters, information about the goodness of fit (log-likelihood, AIC, BIC, correlation between empirical confidence ratings and ratings from a generative model) and trial-by-trial arrays of latent variables (e.g. decision values, metacognitive evidence). The toolbox is highly configurable – in particular, each parameter can be disabled, enabled, or enabled in duplex mode (i.e. sign-dependent, see above).

Parameter fitting minimizes the negative log-likelihood of type 1 choices (sensory level) or type 2 confidence ratings (metacognitive level). For the sensory level, initial guesses for the fitting procedure were found to be of minor importance and are set to reasonable default values. Data are fitted with a gradient-based optimization method (*Sequential Least Squares Programming*; *Kraft, 1988*). However, if enabled, the sensory threshold parameter can introduce a discontinuity in the psychometric function, thereby violating the assumptions of gradient methods. In this case, an additional gradient-free method (*Powell's method*; *Powell, 1964*) is used and the estimate with the lower negative log-likelihood is chosen. Both parameter fitting procedures respect lower and upper bounds for each parameter.

Since parameters of the metacognitive level were found to be more variable, subject-specific initial values for the fitting procedure are of greater importance. For this reason, an initial coarse grid-search with parameter-specific grid points is performed to determine suitable initial values, which are subsequently used for a gradient-based optimization routine (*Sequential Least Squares Programming*). Here too, lower and upper bounds are respected for each parameter.

The toolbox has optional settings to invoke an additional fine-grained grid-search and an explicit global optimization routine (*Basin-hopping*; *Wales and Doye, 1997*), both of which are computationally considerably more expensive. Exploratory tests showed that these methods were not necessary for parameter estimation on either simulated or empirical data in this work; however, this may be different for other empirical datasets.

## Acknowledgements

This research was funded by the German Research Foundation (grant GU 1845/1-1). I'm grateful to the lab of Elisa Filevich for helpful input and critical discussion. Computation has been performed on the HPC for Research cluster of the Berlin Institute of Health.

# Additional information

## Funding

| Funder | Grant reference number | Author |
|---|---|---|
| Deutsche Forschungsgemeinschaft | GU 1845/1-1 | Matthias Guggenmos |

The funders had no role in study design, data collection and interpretation, or the decision to submit the work for publication.

## Author contributions

Matthias Guggenmos, Conceptualization, Resources, Data curation, Software, Formal analysis, Funding acquisition, Validation, Investigation, Visualization, Methodology, Writing - original draft, Project administration, Writing - review and editing

## Author ORCIDs

Matthias Guggenmos ⓘ http://orcid.org/0000-0002-0139-4123

## Decision letter and Author response

Decision letter https://doi.org/10.7554/eLife.75420.sa1
Author response https://doi.org/10.7554/eLife.75420.sa2

# Additional files

## Supplementary files

• Transparent reporting form

## Data availability

The data used for model validation (Shekhar and Rahnev, 2021) was made publicly available by the original authors at https://osf.io/s8fnb/.

The following previously published dataset was used:

| Author(s) | Year | Dataset title | Dataset URL | Database and Identifier |
|---|---|---|---|---|
| Shekhar M, Rahnev D | 2021 | The nature of metacognitive inefficiency in perceptual decision making | https://osf.io/s8fnb/ | Open Science Framework, s8fnb |

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

## Appendix 1

### Derivation of the link function in Equation 5

The link function $c^*(y)$ in **Equation 5** corresponds to an observer who expresses confidence as the subjective probability of having made a correct type 1 decision. Ignoring metacognitive noise and metacognitive biases in a first step, the link function $c^*(y)$ is defined as the (rescaled) choice probability $p$ for the chosen option (rescaled from 0.5..1 to 0..1 using the transformation $2p - 1$). Since the choice probability for the chosen option is symmetric in $y$, the link function can be simplified to just considering absolute decision values (i.e., $c^*(|y|)$). Using the expression for the choice probability in **Equation 7**, a logistic function, and using the relationship $\text{logistic}(x) = 0.5\left(\tanh\left(\frac{x}{2}\right) + 1\right)$, one arrives at the following derivation of the link function:

$$
\begin{aligned}
c^*(|y|) &= 2p\left(\text{choice} = S^+; |y|\right) - 1 = 2\,\text{logistic}\left(\frac{\pi}{\sqrt{3}\sigma_s}|y|\right) - 1 = \\
&= 2 \cdot 0.5\left(\tanh\left(\frac{\pi}{2\sqrt{3}\sigma_s}|y|\right) + 1\right) - 1 = \tanh\left(\frac{\pi}{2\sqrt{3}\sigma_s}|y|\right)
\end{aligned}
\tag{A1}
$$

The final form of the link function in **Equation 5** is based on **Equation A1**, augmented with evidence-based metacognitive bias parameters ($|y| \to z^* := \max\left(\varphi_m |y| + \delta_m, 0\right)$) and accounting for metacognitive readout noise ($z^* \to z$)."

## Appendix 2

**Appendix 2—table 1.** Metacognitive noise distributions.

All distributions are parameterized such that $z^*$ is the mode and $\sigma_m$ is the standard deviation of the distribution (the only exception is the Beta distribution, for which $\sigma_m$ is a spread parameter that cannot be identified with a statistical quantity). For the Gumbel distribution the auxiliary parameter $\eta_m$ was defined as $\eta_m = \pi/(\sigma_m\sqrt{6})$, such that $\sigma_m$ corresponds to the standard deviation of the distribution.

| | *Noisy-readout* | *Noisy-report* |
|---|---|---|
| Censored Normal | $f_m(z) = \begin{cases} \Phi\left(-\frac{z^*}{\sigma_m}\right) & if \quad z=0 \\ \frac{1}{\sigma_m}\phi\left(-\frac{z-z^*}{\sigma_m}\right) & if \quad z>0 \end{cases}$ | $f_m(c) = \begin{cases} \Phi\left(-\frac{c^*}{\sigma_m}\right) & if \quad c=0 \\ \frac{1}{\sigma_m}\phi\left(-\frac{c-c^*}{\sigma_m}\right) & if \quad 0<c<1 \\ \Phi\left(\frac{1-c^*}{\sigma_m}\right) & if \quad c=1 \end{cases}$ |
| Censored Gumbel | $f_m(z) = \begin{cases} 1 - exp\left(-e^{\eta_m z^*}\right) & if \quad z=0 \\ \eta_m exp\left(\eta_m(z-z^*) - e^{\eta_m(z-z^*)}\right) & if \quad z>0 \end{cases}$ | $f_m(c) = \begin{cases} 1 - exp\left(-e^{\eta_m c^*}\right) & if \quad c=0 \\ \eta_m exp\left(\eta_m(c-c^*) - e^{\eta_m(c-c^*)}\right) & if \quad 0<c<1 \\ exp\left(-e^{\eta_m(c^*-1)}\right) & if \quad c=1 \end{cases}$ |
| Truncated Normal | $f_m(z) = \frac{1}{\sigma_m}\frac{\phi\left(-\frac{z-z^*}{\sigma_m}\right)}{1-\Phi\left(-\frac{z^*}{\sigma_m}\right)}$ | $f_m(c) = \frac{1}{\sigma_m}\frac{\phi\left(-\frac{c-c^*}{\sigma_m}\right)}{\Phi\left(\frac{1-c^*}{\sigma_m}\right)-\Phi\left(-\frac{c^*}{\sigma_m}\right)}$ |
| Truncated Gumbel | $f_m(z) = \eta_m \frac{exp\left(\eta_m(z-z^*) - e^{\eta_m(z-z^*)}\right)}{exp\left(-e^{\eta_m z^*}\right)}$ | $f_m(c) = \eta_m \frac{exp\left(\eta_m(c-c^*) - e^{\eta_m(c-c^*)}\right)}{exp\left(-e^{\eta_m c^*}\right)-exp\left(-e^{\eta_m(c^*-1)}\right)}$ |
| Gamma/ Beta | $f_m(z) = \frac{\beta^\alpha}{\Gamma(\alpha)}z^{\alpha-1}e^{-\beta z}$ <br> *Parameterization:* <br> $\alpha = z^{*2} + 2\sigma_m^2 + z^*\sqrt{z^*+4\sigma_m^2}$ <br> $\beta = \frac{1}{2\sigma_m^2}\left(z^* + \sqrt{z^*+4\sigma_m^2}\right)$ | $f_m(c) = \frac{c^{\alpha-1}(1-c)^{\beta-1}}{B(\alpha,\beta)}$ <br> *Parameterization:* <br> $\alpha = c^*\left(\frac{1}{\sigma_m}-2\right)+1$ <br> $\beta = (1-c^*)\left(\frac{1}{\sigma_m}-2\right)+1$ |
| Lognormal | $f_m(z) = \frac{1}{\hat{\sigma}_m^*\sqrt{\pi}}exp\left(-\frac{\ln z - \hat{z}^*}{\hat{\sigma}_m^*}\right)^2$ <br> Note: $\hat{z}^*$ and $\hat{\sigma}_m^*$ represent an analytic parameterization such that the lognormal distribution has mode z* and standard deviation $\sigma_m$. See the published code for details. | |

