## [Editor Report]

This paper presents a novel computational model of metacognition and a validated toolbox for fitting it to empirical data. By formalizing different sources of noise and bias that impact confidence, the proposed model aims at providing metacognition metrics that are independent of perception – a continued endeavor in the field. The framework and toolbox constitute a valuable resource for the field.

---

## [Decision Letter]

**Decision letter after peer review:**

Thank you for submitting your article "Reverse engineering of metacognition" for consideration by *eLife*. Your article has been reviewed by 2 peer reviewers, and the evaluation has been overseen by Valentin Wyart as the Reviewing Editor and Michael Frank as the Senior Editor. The following individual involved in review of your submission has agreed to reveal their identity: Steve Fleming (Reviewer #2).

The reviewers have discussed their reviews with one another, and the Reviewing Editor has drafted this to help you prepare a revised submission. As you will see, the reviewers have found your modeling approach to the measure of metacognition to be interesting and potentially insightful, but several additional analyses and controls will need to be performed for the article to be considered as publishable in *eLife*. Please address all essential revisions below in a revised version of the article, together with a point-by-point response. The individual reviews from the two reviewers are appended below, but they do not formally require individual point-by-point responses at the revision stage.

Essential revisions:

1) Parameter and model recovery: separability between the two metacognitive modules. More work needs to be done to demonstrate that the proposed model can discriminate between the noisy readout module and the noisy report module. The two proposed modules have different psychological meanings, but seem to impact the confidence output similarly. Indeed, qualitatively, it seems like the only thing distinguishing them is that the noise is either applied before or after the link function, and it isn't clear whether this was sufficient to distinguish one from the other. Are these two modules mutually exclusive (as Figure 1 suggests), or could both sources of noise co-exist? It is important to show model recovery for introducing noisy readout vs. report at the metacognitive level. Both reviewers appreciate they might return differential AICs, but it is important to report a 2x2 model confusion matrix from simulated data (see Wilson and Collins, 2019 *eLife*) to test whether the ground-truth metacognitive module can be recovered from simulated data. The similarity between the two metacognitive modules also raises the question of how the two types of σ_m are recoverable/separable from each other. If they capture independent aspects of noise, one could imagine a model with both modules. More evidence is needed to show that these two capture separate aspects of noise.

2) Parameter and model recovery: perform analyses that capture more realistically aspects of experimental datasets. The parameter recovery demonstrated in Figure 4 is impressive, but it is critically important to know what happens when more than one parameter needs to be inferred, as in real data. The plots don't show what the other parameters are doing when one is being recovered (nor do the plots in the supplement to Figure 6). The key question is whether each parameter is independently identifiable, or whether there are correlations in parameter estimates that might limit the assignment of effects (e.g., metacognitive bias) to one parameter rather than another. For example, the slope and metacognitive noise may trade off against each other, as might the slope and δ_m. This seems particularly important to establish as a limit of what can be inferred from a ReMeta model fit. To address this concern, a proper correlation matrix between best-fitting parameters should be presented, and a parameter confusion matrix should be conducted across the parameter space, not only for certain regimes (i.e. more than Figure 6 supp 3), that is, the full grid exploration irrespective of how other parameters were set. Finally, recovery analyses should not (only) be done on 10,000 trials which is one to two orders of magnitude larger than the amount of data collected from individual subjects in experiments. 1,000 trials appear like an upper bound on typical data.

3) Trade-off between the flexibility of the model vs. the generalizability of the identified metacognitive architecture across contexts and participants. The current modeling framework proposed appears to favor flexibility (reflected, e.g., in the modularity of the metacognitive part, choice of the link functions) against the generalizability of the identified architecture. But beyond questions about model and parameter recovery that need to be taken care of, could the modeling framework be 'too flexible' in that it does not allow to draw conclusions that generalize across contexts (e.g., cognitive tasks, stimuli, etc.) and participants. This question is important, because Figure 7 and ‘Application to empirical data’ of the results explain that all models are similar, regardless of module of functions specified; Figure 7 supp shows that half of participants are best fitted by noisy readout, while the other half is best fitted by noisy report; plus, idiosyncrasies across participants are all captured. It would therefore be important to discuss in the article whether the high flexibility of the modeling architecture (that captures idiosyncrasies using its various free architectural choices and parameters) may compromise the generalizability of the modeling results at the group level and across tasks. This will be important to understand better the strengths and possible weaknesses of the proposed modeling framework for metacognition.

4) Separate fitting of type-1 and type-2 stages. The final paragraph of the discussion explains that data on empirical trial-by-trial accuracy is not used in the model fits. It is easy to see how in a process model that simulates decision and confidence data from stimulus features (from the perspective of the modeled observed), objective accuracy should not be considered as an input. But in terms of a model fit, it seems odd not to use trial by trial accuracy to constrain the fits at the metacognitive level, given that the hallmark of metacognitive sensitivity is a confidence-accuracy correlation. Is it not possible to create accuracy-conditional likelihood functions when fitting the confidence rating data (similar to how the meta-d' model fit is handled)? Psychologically, this also makes sense given that the observer typically knows their own response when giving a confidence rating. It is very important to explain more explicitly why fitting both choices and confidence at the same time is not possible in the current modeling framework. The assumption that different sources of noise are independent does not appear sufficient to explain this modeling choice.

5) Differences in the tasks required to fit the ReMeta model and the Mration model. An important nuance in comparing the present σ_m to Mratio is that the present model requires that multiple difficulty levels are tested, whereas instead, the Mratio model based on signal detection theory assumes a constant signal strength. How does this impact the (unfair?) comparison of these two metrics on empirical data that varied in difficulty level across trials? Relatedly, the Discussion paragraph that explained how the present model departs from type 2 AUROC analysis similarly omits to account for the fact that studies relying on the latter typically intend to not vary stimulus intensity at the level of the experimenter.

6) Structure of the model: variability in scale usage. Variability in scale usage appears to be forced to be set early in the model, not late. This is concerning that all the variability in scale usage is being assumed to load onto evidence-related parameters (eg δ_m) rather than being something about how subjects report or use an arbitrary confidence scale (eg the "implicit biases" assumed to govern the upper and lower bounds of the link function). You could have a similar notion of offset at the level of report – eg an equivalent parameter to δ_m but now applied to c and not z. Would these be distinguishable? They seem to have quite different interpretations psychologically: one is at the level of a bias in confidence formation, and the other at the level of a public report.

7) Structure of the model: integration only of choice-congruent decision evidence for confidence. In Eq8, could you explain why only the decision values consistent with the empirical choice are filtered. Is this an explicit modeling of the 'decision-congruence' phenomenon reported elsewhere (eg. Peters et al. 2017; Luu and Stocker, 2018, *eLife*)? What would be the implications of not keeping only the congruent decision values? It is important to motivate more clearly and explicitly this choice in the structure of the model.

8) Structure of the model: λ_m. It appears that λ_m is a meaningful part of the model. If so, it should be introduced early into the Figure 1 model, and be properly part of the parameter recovery procedure described above.

*Reviewer #1 (Recommendations for the authors):*

I did not have time to check the toolbox available online but I note that it is an important strength that the authors have shared this resource for other researchers to look at or re-use for their own work.

Regarding the reasoning in paragraph 1.6, it is unclear to me why metacognitive evidence for the chosen option would become zero in case of a sign flip, rather than becoming negative evidence (just flipping sign)? I think it would be best to simply make the assumption that sign flips are impossible.

Isn't the lack of a reliable recovery of δ_m at low and high type 1 performance levels an issue, because it is exactly at the bounds that δ_m is supposed to have an effect?

We would like to see more discussion on how this model compares to other proposals of Bayesian confidence signatures (Adler and Ma, 2018, already cited). I also wondered about the possible inclusion of RTs in the model, which is then nicely addressed in the Discussion already.

Figure 4, middle panels: I think it is an assumption to simply convert confidence in 0-1 space to 0.5-1 space. Indeed, observers may treat very differently a 0.5-1 scale in which both 'I have purely guessed' and 'I am pretty sure I have made an error' would be reported around 0.5, whereas would be further apart on a 0-1 scale.

The sensory bias (bias), sensory noise (slope), and sensory threshold (random responses) all capture choice patterns in a logistic function; can you better explain how Equation 2 was developed? But parameterization of Figure 2 seems able to capture all standard effects. Similarly the reasoning leading to the generation of Equation 5 could be better motivated.

Figure 3C legend "Higher metacognitive noise flattens the relationship between type 1 decision values and confidence.": this is between metacognitive evidence and confidence instead?

The behavioral effects shown in Figure 2 and 3 as a function of parameter values are useful, but also confusing because several of the parameters change value from plot to plot. Would it be possible instead to fix all but one parameter, and change the one parameter for 4-5 values instead of 2 values, for instance using a color scale? This way, the reader would be able to appreciate the effect of each parameter in isolation from the others.

Figure 6A displays an increase in Mratio as type 1 d' increases – the opposite of what is reported in the legend and in the text? at least for d' between 0 and 3, which is the case in most perceptual experiments? Likewise, there is a discrepancy with σ_m from the other module (Figure 6 supp).

*Reviewer #2 (Recommendations for the authors):*

- I found it odd that z was the noisy estimate of z-hat (and c the noisy estimate of c-hat), rather than the other way around given that the -hat operator is typically added to refer to an estimate.

- The current model is restricted to cases in which the sensory evidence is varying. This is opposite to the meta-d' model, in which sensory evidence is assumed to be fixed, or at least varying across a narrow range (eg d' is constant for stimulus repetitions). It might be worth emphasising that the two models can be chosen depending on the data available, rather than ReMeta being universally more suitable than meta-d'.

- I felt the introduction could do with some more emphatic framing, and that the author is selling himself short here. Lines 26-33 outline the rationale for the model. But there are two goals here - one is an incremental one of fixing the biases in current metacognitive efficiency estimates, which is useful, but it doesn't seem to be so debilitating (at least with the standard m-ratio estimates) as to warrant entirely new model machinery. But then later in the paragraph, the fact that this new approach could also accommodate fits of parameters governing different types of metacognitive biases is introduced. This seems much more important given that there is no current framework for modelling such biases.

---

## [Author Response]

Reviewer #1 (Recommendations for the authors):I did not have time to check the toolbox available online but I note that it is an important strength that the authors have shared this resource for other researchers to look at or re-use for their own work.Regarding the reasoning in paragraph 1.6, it is unclear to me why metacognitive evidence for the chosen option would become zero in case of a sign flip, rather than becoming negative evidence (just flipping sign)? I think it would be best to simply make the assumption that sign flips are impossible.

Indeed, re-reading this paragraph I found my wording to be unnecessarily convoluted. The point I had in mind is quite straightforward: either sign flips are impossible due to the nature of metacognitive noise itself (e.g. lognormal distribution) or they are possible but are not observed because the confidence scale does not include the possibility to report errors (hence confidence=0 in such cases -> censored distributions). I substantially simplified the corresponding paragraphs along these lines (‘Metacognitive noise: noisy-report models’).

Isn't the lack of a reliable recovery of δ_m at low and high type 1 performance levels an issue, because it is exactly at the bounds that δ_m is supposed to have an effect?

Figure 4 (second row) shows that the recovery of δ_m indeed becomes unstable at very low or very high type 1 performance levels. I don’t consider this problematic, however.

Figure 4 investigates parameter recovery in dependence of *overall* type 1 performance. As outlined above, if overall type 1 performance is close to chance or close to perfect, behavior is random or shows little variance, respectively, which is why parameter recovery is often hampered.

More to the reviewer’s point, in the manuscript I provide an interpretation of δ_m in terms of a confidence threshold (for δ_m < 0), i.e. a minimal level of sensory evidence required to have a nonzero confidence experience. I assume this is what the reviewer was referring to with “exactly at the bounds that δ_m is supposed to have an effect”; please correct me otherwise. This interpretation, however, refers to instances of *single* trials in which sensory evidence is low (from the perspective of the observer, not necessarily objectively). Critically, the idea of a confidence threshold can be meaningful and impactful even if overall performance is at intermediate or high levels, as subjective sensory evidence will often nevertheless be low in a *certain fraction* of trials.

More importantly, however, the evidence shift induced through δ_m applies to all levels of internal evidence (after all, it is just the subtraction of a constant); the idea of a confidence threshold at very low levels of evidence is highlighted mainly because it is associated with a prominent feature in the confidence-evidence relationship.

We would like to see more discussion on how this model compares to other proposals of Bayesian confidence signatures (Adler and Ma, 2018, already cited). I also wondered about the possible inclusion of RTs in the model, which is then nicely addressed in the Discussion already.

As the reviewer mentions, I had cited a paper by Adler and Ma from 2018 (*Neural Computation*), but I now realized that there is a second Adler and Ma (2018; *PLOS Comp. Biology*), to which the reviewer is likely referring to. I had missed the latter one in my literature review. I now refer to this and related references in a new discussion paragraph on Bayesian confidence models (Line 807ff):

“Finally, how does the present model relate to the recent discussion between Bayesian and Non-Bayesian models of confidence (Aitchison et al., 2015; Sanders et al., 2016; Adler and Ma, 2018b)? A Bayesian observer of the (inner) world is one who maintains a posterior probability density over possible states of that world. In particular, computing confidence for such an observer corresponds to integrating the posterior over all possible states for which the type 1 choice would be correct. In this sense, the model proposed here with the link function provided in Equation 5 corresponds to a Bayesian observer, albeit one that can be susceptible to metacognitive biases and to additional sources of metacognitive noise. Thus, while the observer is Bayesian in nature, it may not be Bayes optimal. At the same time, the framework and the toolbox are flexible to allow for “non-Bayesian” link functions (Figure 3—figure supplement 1) that could represent certain idiosyncratic heuristics and shortcuts inherent to human confidence judgements. Of note, the model proposed here does not consider prior distributions over the stimulus categories (see e.g., Adler and Ma, 2018b). Instead, it is assumed that the observer considers both stimulus categories equally likely which is a reasonable assumption if stimulus categories are balanced.”

I agree that including RTs in a confidence model would be a nice feature, but in my opinion this requires a lot of groundwork that is beyond the scope of this work.

Figure 4, middle panels: I think it is an assumption to simply convert confidence in 0-1 space to 0.5-1 space. Indeed, observers may treat very differently a 0.5-1 scale in which both 'I have purely guessed' and 'I am pretty sure I have made an error' would be reported around 0.5, whereas would be further apart on a 0-1 scale.

In this manuscript I strictly consider confidence as ranging from ‘I have purely guessed’ to ‘I am 100% certain’, i.e. I do not consider the case of realizing errors at the time of the confidence report. This was stated e.g. on Line 330ff (“Unless confidence rating scales include the possibility to indicate errors (which I do not consider here)[.]”). The transformation from 0.5-1 to 0-1 space is thus a purely mathematical one, motivated by certain technical advantages (e.g. the Β noise distribution is naturally bounded between 0 and 1). I now also state this in the relevant paragraph concerning the transformation 0.5-1 -> 0-1 (Line 183ff):

“Note that I do not consider the possibility that type 1 errors can be reported at the time of the confidence report, i.e., confidence cannot be negative.”.

The sensory bias (bias), sensory noise (slope), and sensory threshold (random responses) all capture choice patterns in a logistic function; can you better explain how Equation 2 was developed? But parameterization of Figure 2 seems able to capture all standard effects. Similarly the reasoning leading to the generation of Equation 5 could be better motivated.

Equation 2: The formula in Equation 2 is the logistic distribution. The only change from the standard form is that I converted the conventional parameter *s* to a standard deviation *σ* using fact that the variance of the logistic distribution is known as *s*²*π*²/3. The nature of the bias parameter in Equation 1 corresponds to a horizontal shift of the resulting psychometric function. The sensory threshold parameter is the mathematical formalization of the notion that a certain degree of sensory stimulation is necessary to drive the system, i.e., below a certain intensity level *δ*_s_ the resulting decision values are zero. I now provide this explanatory information interspersed in ‘Computing decision values’.

Equation 5: I have now added the derivation of the link function in Equation 5 as Appendix Equation A1 and reference to it in ‘The link function: from metacognitive evidence to confidence’.

Figure 3C legend "Higher metacognitive noise flattens the relationship between type 1 decision values and confidence.": this is between metacognitive evidence and confidence instead?

Thanks, corrected!

The behavioral effects shown in Figure 2 and 3 as a function of parameter values are useful, but also confusing because several of the parameters change value from plot to plot. Would it be possible instead to fix all but one parameter, and change the one parameter for 4-5 values instead of 2 values, for instance using a color scale? This way, the reader would be able to appreciate the effect of each parameter in isolation from the others.

I liked this suggestion and implemented it for Figures 2 and 3:

Figure 6A displays an increase in Mratio as type 1 d' increases – the opposite of what is reported in the legend and in the text? at least for d' between 0 and 3, which is the case in most perceptual experiments? Likewise, there is a discrepancy with σ_m from the other module (Figure 6 supp).

Thanks for noting. I replaced it with a more neutral “shows a nonlinear dependency with varying type 1 performance levels” (Line 387). Note that the plots in Figure 6 changed slightly because I now plot proportion correct responses instead of d’ and I use truncated normal distributions for all plots (which is the new default of the toolbox; also, it makes the comparison between noisy-readout and noisy-report models easier).

Reviewer #2 (Recommendations for the authors):- I found it odd that z was the noisy estimate of z-hat (and c the noisy estimate of c-hat), rather than the other way around given that the -hat operator is typically added to refer to an estimate.

I agree that the notation could be confusing. I now replaced the hat-notation with an asterisk-notation. I did not simply flip the hat and non-hat notation, since noisy versions of the variables are not really an estimate in the traditional sense either (as e.g., the sample mean).

- The current model is restricted to cases in which the sensory evidence is varying. This is opposite to the meta-d' model, in which sensory evidence is assumed to be fixed, or at least varying across a narrow range (eg d' is constant for stimulus repetitions). It might be worth emphasising that the two models can be chosen depending on the data available, rather than ReMeta being universally more suitable than meta-d'.

As I noted also to Reviewer #1, this restriction was unnecessarily imposed in the previous version of the manuscript. The references to this restriction are now removed from the manuscript. In other words, the model also works for constant stimuli.

- I felt the introduction could do with some more emphatic framing, and that the author is selling himself short here. Lines 26-33 outline the rationale for the model. But there are two goals here - one is an incremental one of fixing the biases in current metacognitive efficiency estimates, which is useful, but it doesn't seem to be so debilitating (at least with the standard m-ratio estimates) as to warrant entirely new model machinery. But then later in the paragraph, the fact that this new approach could also accommodate fits of parameters governing different types of metacognitive biases is introduced. This seems much more important given that there is no current framework for modelling such biases.

I agree with this assessment and I now put a stronger emphasis on this methodological gap in the literature (Line 53ff):

“However, currently there is no established framework that allows for unbiased estimates of metacognitive biases. The validity of traditional calibration curve analyses, which is based on a comparison of the subjective and objective probability of being correct, has been debunked repeatedly (Soll, 1996; Merkle, 2009; Drugowitsch, 2016). In particular, the classic hard-easy (Lichtenstein and Fischhoff, 1977), according to which overconfidence is particularly pronounced for difficult tasks, can be explained as a mere statistical artefact of random errors. For this reason, and in view of the potential importance in patient populations, there is a pressing need for unbiased measures of metacognitive biases.”

Towards the end of the introduction, I once again refer to this point (Line 111ff):

“[.] As outlined above, there is currently no established methodology to measure under- and overconfidence, let alone measure different types of such biases. [..]”

In return, I cut down on introductory space taken up by the issue of metacognitive efficiency, in line also with the recommendation of Reviewer #1.